# The Underlying Mechanisms and Emerging Strategies to Overcome Resistance in Breast Cancer

**DOI:** 10.3390/cancers17172938

**Published:** 2025-09-08

**Authors:** Krishnaswamy Kannan, Alagarsamy Srinivasan, Aarthi Kannan, Nawab Ali

**Affiliations:** 1Biomolecular Integrations, Little Rock, AR 72205, USA; krishna.kannan9900@gmail.com; 2Nanobio Diagnostics, West Chester, PA 19382, USA; alagarsamy.srinivasan@gmail.com; 3Department of Dermatology, University of California-Irvine, Irvine, CA 92697, USA; 4Department of Biology, Donaghey College of Science, Technology, Engineering, and Mathematics, University of Arkansas at Little Rock, Little Rock, AR 72204, USA

**Keywords:** breast cancer, tumor resistance, AI, immunotherapy, experimental models, TNBC, triple-negative breast cancer, apoptosis, regulated cell death, tumor microenvironment, bone metastasis, microbiota

## Abstract

Breast cancer (BC) remains a significant global health challenge, partly due to its ability to develop resistance to treatment. BC resistance arises from various biological mechanisms, including cancer cells actively expelling drugs, altering their internal metabolic and signaling pathways, evading the immune system, or persisting as drug-tolerant cancer stem cells. These changes occur in the context of a tumor microenvironment that further shields the cancer, limiting treatment access. As the disease advances, particularly when it spreads to other body parts, effective treatment becomes increasingly difficult. Recent advancements in artificial intelligence (AI) offer promising avenues for improving early detection, refining diagnoses, enhancing clinical decision-making and thereby personalizing treatment strategies. This review examines the biological mechanisms that cause BC resistance and its treatment. It further explores AI’s potential to address critical challenges, transforming BC therapy to provide improved survival rates and enhanced quality of life for patients.

## 1. Introduction

Breast cancer (BC) remains a paramount global health challenge and a leading cause of cancer-related morbidity and mortality worldwide. Globally, approximately 2.3 million new female cases were diagnosed, and 670,000 deaths were reported for the year 2022 [1,2,3]. The BC incidence increases with age; the median age of early diagnosis among women in the US is 62 years. It is projected that the global incidence of BC is likely to increase 40% by the year 2040 [4]. The American Cancer Society projects approximately 316,950 new cases of invasive BC and 59,080 cases of ‘ductal carcinoma in situ’ (DCIS) for the year 2025 [5,6].

Despite the significant advances in early detection and adjuvant therapies that have improved outcomes for localized disease, the prognosis for metastatic breast cancer (MBC) remains poor. Notably, metastasis is the primary driver of BC-related mortality, accounting for approximately 90% of deaths, highlighting the failure of treatment [3,6]. Within the BC subtypes, triple-negative breast cancer (TNBC) is particularly aggressive, with a median survival of only 13.3 months [7]. The current challenge is the frequent development of ‘therapeutic resistance’, a complex biological process wherein cancer cells adapt and evolve to evade the effects of anticancer treatments. After the initial favorable responses to endocrine therapy and chemotherapy, tumors often acquire resistance, leading to disease progression, recurrence, and diminished patient survival [8,9,10].

Given the impact of therapeutic resistance on the clinical outcomes in BC, a better understanding of the mechanisms underlying resistance is paramount to develop effective therapeutic strategies [11,12,13]. Resistance develops through the interplay of tumor-intrinsic heterogeneity and tumor-extrinsic influences, including the tumor microenvironment and immune–metabolic interactions. While Xiong et al. (2025) [14] and Dhiman et al. (2024) [15] offered broad overviews of BC pathophysiology and nanocarrier applications, this review delivers a more granular analysis of various resistance mechanisms reported in the literature.

This review systematically evaluates the resistance mechanisms in breast cancer (BC), emphasizing a central hub where various factors converge. These factors include genetic and epigenetic alterations, dysregulated signaling, metabolic rewiring, regulated cell death, microbial influences, cancer stem cell plasticity, and exosome-mediated communication. Particular attention is paid to the metabolic–immune axis and the tumor microenvironment (TME), which act as critical drivers of metastatic progression and therapeutic resistance. In parallel, this review also focuses on the role of AI in BC resistance. AI-enabled approaches enhance early diagnosis and patient-tailored therapeutic strategies by analyzing patient data on circulating tumor DNA (ctDNA), advanced imaging, and prediction of efflux and mutation-associated resistance. Together, these insights underscore how AI can illuminate the complex interplay between metabolism, immunity, and the TME, offering new opportunities to overcome resistance and improve outcomes in BC [11,16,17].

### 1.1. Biological Characteristics of Breast Cancer

BC originates from the epithelial lining of mammary ducts or lobules. Histologically, it is predominantly classified as ductal or lobular carcinoma [18,19]. Invasive ductal carcinoma is the most common subtype, accounting for 70–80% of all BC cases. The molecular classification of BC is primarily based on the expression of hormone receptors (estrogen receptor [ER], progesterone receptor [PR]) and human epidermal growth factor receptor 2 (HER2) [20,21]. Approximately 70% of BCs are hormone-receptor-positive (HR^+^), encompassing molecular subtypes such as luminal A and luminal B (both HR^+^), HER2-enriched, and TNBC. Each of these subtypes exhibits distinct molecular characteristics and therapeutic vulnerabilities [22]. Notably, the ‘HER2-low’ expressing subtype has recently emerged as a clinically relevant entity [23], warranting further investigation. Conversely, TNBC, constituting 10–15% of cases, is defined by the absence of HER2, ER, and PR expression. Due to its aggressive biological behavior, treatment for TNBC primarily relies on systemic chemotherapy regimens (e.g., anthracyclines, taxanes), although recent advances in immunotherapy and combination therapies have demonstrated promising outcomes [24]. Hormone-receptor-positive tumors (70%) are characterized by their dependence on estrogen and/or progesterone signaling pathways and are effectively treated with endocrine therapies, such as tamoxifen and aromatase inhibitors [25].

The goals of our review are to highlight BC pathogenesis, treatment and recurrence of the disease in the context of potential resistance mechanisms. As multiple factors contribute to resistance, it is likely that an update on this subject may serve to uncover avenues for treatment. A systematic review of the resistance mechanisms in the BC subtypes, including TNBC, was conducted. PubMed, Scopus, Web of Science, EMBASE, and Cochrane Library were searched for peer-reviewed, English-language articles, supplemented by manual searches and cross-referencing of medical journals. Boolean operators (AND, OR, NOT), combined keywords and controlled vocabularies (e.g., MeSH, EMTREE) were used to ensure comprehensive retrieval. Multiple authors were involved in screening and assessing full-text articles.

### 1.2. Genetic Risk Factors

Before discussing resistance-related genetic changes, it is important to outline the inherited mutations that increase the BC risk. Germline mutations in the *BRCA1* and *BRCA2* genes significantly elevate the lifetime BC risk—up to 70% by age 80—and are often linked to earlier onset and bilateral disease [26]. Mutations in *PALB2* confer a 14% risk by age 50 and up to 35% by age 70 [27]. Carriers of inactivating mutations in the *ATM* gene face a twofold increase in risk. Other moderate-risk genes include *BARD1*, *RAD51C*, and *RAD51D*, which are associated with ER-negative BC, while *CDH1* and *CHEK2* gene mutations are more often linked to ER-positive disease [21]. Recognizing these genetic predispositions, the National Comprehensive Cancer Network (NCCN) recommends genetic counseling and testing for high-risk individuals to guide early detection and preventive strategies.

### 1.3. Diagnosis and Progression

BC diagnosis typically follows a three-step process: (a) evaluation of medical history and clinical breast examination, (b) imaging via diagnostic mammography or ultrasound, and (c) histological analysis of biopsy specimens. Furthermore, magnetic resonance imaging (MRI) is a valuable tool for diagnosing and managing BC. Breast MRI boasts high sensitivity in detecting BC and is a major advantage for women with dense breasts where mammography might be less effective [source: www.hopkinsmedicine.org, (accessed on 29 August 2025)]. To minimize artifacts and misdiagnosis, Li et al. [28] have developed the TabPFN algorithm, which has increased the sensitivity of traditional MRI.

Localized disease is classified as in situ when malignant cells remain confined within the ductal or lobular basement membrane. Invasion occurs when these cells breach the membrane and infiltrate surrounding tissue [18,25]. Metastatic BC involves distant spread, commonly to the lungs, liver, bones, and brain, and requires comprehensive treatment strategies. These may include systemic chemotherapy, radiotherapy, and targeted immunotherapies, such as immune checkpoint inhibitors, depending on the tumor’s molecular subtype (see Figure 1). The current trend in artificial intelligence (AI) is increasingly advancing the accuracy of histopathological classification and diagnosis and enhancing the treatment response [29]. Machine learning algorithms now achieve sensitivity and specificity rates exceeding 90% in mammographic image interpretation—often outperforming human radiologists [30].

### 1.4. Metastasis and Disease Complexity

Metastatic BC accounts for nearly 90% of all BC-related deaths, underscoring its clinical urgency [3,31]. The complexity arises from an interplay of chronic inflammation, immune suppression, and organ-specific factors that collectively create a supportive niche for tumor cell colonization and growth [11]. Emerging studies in cancer neuroscience suggest that tumor cells may exploit neuro-immune signaling and neural remodeling to facilitate metastasis and immune evasion [32]. Several established risk factors increase the likelihood of developing BC. Advancing age is linked to cumulative DNA damage and declining repair mechanisms [33]. A family history, especially of *BRCA1* or *BRCA2* gene mutations, confers a high hereditary risk [34]. Women previously diagnosed with BC are at greater risk of developing contralateral tumors [33]. Early menarche (before age 12) and late menopause (after age 55) prolong hormonal exposure, increasing risk [33]. Nulliparity or first childbirth after age 30 is associated with hormonal imbalance and delayed breast tissue maturation [35]. High breast density, marked by excess glandular tissue, is another independent risk factor and complicates mammographic detection [33].

Lifestyle factors, including alcohol, obesity, and smoking, contribute through increased estrogen, inflammation, and exposure to carcinogens [36]. Hormone replacement therapy, especially estrogen–progestin combinations, also raises the risk by elevating circulating hormones [33]. Radiation exposure during youth can damage breast DNA and elevate the long-term risk [33]. Lastly, utero exposure to diethylstilbestrol (DES) increases the lifetime risk for both exposed women and their daughters [33,37].

### 1.5. Bioinformatics Approaches to BC Resistance

Beyond epidemiological risk factors, bioinformatics analyses have revealed molecular determinants of breast cancer susceptibility and resistance. TCGA and METABRIC studies have catalogued recurrent alterations in *BRCA1*, *BRCA2*, *TP53*, *PIK3CA*, and *ESR1*. These findings link family history and hormonal exposures to well-defined mutational landscapes [38]. Transcriptomic datasets such as GEO and ArrayExpress have identified signatures of proliferation (Ki-67, CCND1), immune evasion (PD-L1/CD274, CTLA4), and metabolic rewiring (HK2, LDHA) [39]. These expression profiles help stratify high-risk subgroups. Proteogenomic studies, including CPTAC, highlight dysregulation of PI3K/AKT/mTOR and MAPK signaling, as well as DNA damage–repair networks [40]. Such alterations connect obesity and radiation exposure to tumor aggressiveness [41]. Epigenomic analyses reveal promoter hypermethylation of *ESR1* and *BRCA1*, as well as regulatory roles for non-coding RNAs such as MIR21 (miR-21) and long non-coding RNA-*HOTAIR* [42,43]. Survival-linked meta-analyses using cBioPortal and Kaplan–Meier Plotter show that high expression of stemness markers (ALDH1A1, SOX9) and EMT drivers (TWIST1, SNAI2) predicts poor outcomes and therapeutic resistance. Some of the major findings on BC pathogenesis relating to bioinformatics are presented in Table 1. Together, these bioinformatics findings complement the literature-based risk factors and provide a mechanistic framework for resistant disease states.

## 2. Early Detection and Diagnostic Technologies

Early detection of BC is critical for improving outcomes. Cutting-edge technologies, including high-resolution imaging, molecular profiling, and liquid biopsies, are transforming diagnostic precision and resistance monitoring [44]. Non-invasive tools like circulating tumor DNA (ctDNA) assays, circulating tumor cells (CTCs), and exosomal biomarkers show promise for early detection. These tools help with real-time disease tracking and predicting resistance, especially in aggressive subtypes like TNBC [45,46,47]. However, challenges such as resistance mechanisms in advanced or heterogeneous tumors continue to hinder therapeutic success.

Emerging solutions such as AI-driven diagnostics, multi-omics integration, and personalized treatment planning offer hope but face barriers to widespread adoption, including standardization, sensitivity, and equitable access [48,49,50]. These advancements herald a shift toward precision oncology, though rigorous clinical validation and thoughtful integration into practice are vital to unlock their full potential.

## 3. Therapeutic Strategies and Ongoing Challenges

BC treatment involves a multimodal approach, including surgery, radiotherapy, chemotherapy, endocrine therapy, and targeted agents [25]. Surgical management of BC has evolved toward precision and de-escalation: breast-conserving procedures (lumpectomy, wide excision, quadrantectomy) are now preferred in early-stage tumors, typically followed by radiotherapy to reduce recurrence [51]. Oncoplastic techniques enhance cosmetic outcomes without compromising control. Recent trials, including SOUND (Sentinel Node vs. Observation After Axillary Ultra-Sound) [52] and INSEMA (Comparison of Axillary Sentinel Lymph Node Biopsy Versus no Axillary Surgery) [53], support further omission of axillary interventions in selected low-risk patients. Multidisciplinary evaluation now guides surgical planning, balancing tumor biology, systemic therapy timing, and quality-of-life outcomes.

For hormone-receptor-positive (HR^+^) cancers, endocrine therapies such as tamoxifen and aromatase inhibitors remain the standard [54]. Targeted therapies like ‘trastuzumab’ are essential in HER2-positive BC. In postmenopausal HR^+^ patients, extended aromatase inhibitor therapy has shown superior outcomes to tamoxifen [55]. Immunotherapy, particularly ‘pembrolizumab’, has shown benefit in high-risk TNBC [56].

In metastatic BC, systemic therapies dominate. Options include chemotherapy (e.g., capecitabine), endocrine therapy combined with CDK4/6 inhibitors (e.g., palbociclib) for HR^+^ disease, ‘trastuzumab’ for HER2^+^ tumors, and PARP inhibitors (e.g., olaparib) for BRCA-mutant cancers [57]. Immunotherapy benefits PD-L1+ TNBC, while ‘denosumab’ helps manage bone metastases, often alongside palliative care [58]. Treatment selection is guided by the molecular subtype (HR^+^, HER2^+^, TNBC), patient comorbidities, and genomic markers. Clinical trials continue to explore antibody–drug conjugates (ADCs) and personalized cancer vaccines. HER2-directed regimens (e.g., trastuzumab + pertuzumab) have improved the outcomes in HER2^+^ BC [59]. CDK4/6 inhibitors (palbociclib, ribociclib, abemaciclib) have significantly prolonged progression-free survival in HR^+^/HER2− MBC [60]. PARP inhibitors show efficacy in BRCA-mutated BC. In TNBC, combining ‘pembrolizumab’ with chemotherapy has improved the pathological complete response in neoadjuvant settings [61].

Advances in early detection and targeted treatments have raised the five-year survival rate for localized BC to nearly 90% [6]. Molecular profiling and AI-enhanced diagnostics enable personalized treatment strategies, reducing toxicity and improving outcomes [44,62]. Endocrine and HER2-targeted therapies have decreased recurrence in the HR^+^ and HER2^+^ subtypes. However, therapeutic resistance remains a major hurdle, often driven by somatic mutations and tumor heterogeneity [11,25,63]. TNBC remains particularly challenging due to its aggressive nature and limited targeted options, though immunotherapy offers new hope [34]. Long-term toxicities such as cardiotoxicity from anthracyclines and bone loss from aromatase inhibitors continue to affect survivorship [64,65]. Additionally, disparities in access to advanced treatments hinder equitable care. Addressing resistance, improving tolerability, and ensuring global access remain critical goals for advancing BC therapy.

## 4. Resistance Mechanisms in Breast Cancer

BC resistance to conventional therapies remains a significant obstacle to achieving durable remissions and cures across all metastatic subtypes, such as hormone-receptor-positive (HR^+^), HER2-positive, and triple-negative (TNBC) [8,10,34,66]. While intricate molecular and cellular resistance mechanisms have been delineated, innovative strategies targeting metabolic vulnerabilities, epigenetic alterations, immune dynamics, and computational advancements are emerging as countermeasures. This section reviews the underlying mechanisms of BC resistance and highlights novel therapeutics and technologies aimed at overcoming them. The mechanisms emphasized include cancer metabolism, mitochondrial function, cancer stem cells (BCSCs), the TME, immunometabolism, immunotherapies, and single-cell RNA sequencing (scRNA-seq). Although these barriers pose major challenges, emerging strategies provide opportunities for overcoming them [8,10,34,66]. Recent advances in AI medicine aid in early detection, whereas nanotechnology enables targeted drug delivery [62]. Collectively, these approaches aim to disrupt resistance pathways and enhance treatment efficacy across BC subtypes and ultimately improve patient outcomes.

### 4.1. Experimental Models of Resistance Mechanisms

The experimental models are indispensable, offering controlled systems to study drug resistance, tumor–stromal interactions, and genetic or epigenetic drivers. Unlike clinical practice, where access to patient tumor samples at resistance onset is limited, preclinical models allow longitudinal, mechanistic studies to detect resistance pathways earlier and test interventions [67]. However, species differences between mouse and human tumors limit extrapolation. As technologies evolve, including CRISPR/Cas9 gene editing, humanized mouse models, 3D bioprinting, and patient-derived organoids, the gap between preclinical findings and clinical applications is becoming narrower, if not totally absent [67,68,69]. Thus, many experimental models are applicable to the human situation and enable drug discovery [69].

Over the years, a spectrum of experimental models has been developed and refined to capture the complexity of BC resistance. These include conventional 2D cell line models, more physiologically relevant 3D spheroids and organoids [67,68]. Additionally, patient-derived xenografts (PDXs), genetically engineered mouse models (GEMMs), circulating tumor cell (CTC)-derived models, and in silico computational models help advance BC research [67,68,69,70]. Each of these models offers distinct insights. The 2D models have illuminated estrogen receptor (ER) signaling, HER2 amplification, and ABC transporter-mediated drug efflux. The organoid and spheroid systems better replicate the spatial architecture and TME-induced drug gradients observed in vivo.

As technologies evolve, CRISPR/Cas9 gene editing, humanized mouse models, 3D bioprinting, and patient-derived organoids have narrowed, though not eliminated, the gap between preclinical and clinical applications [66,67,68]. Thus, many experimental models now translate more effectively to human disease and enable drug discovery [68].

Importantly, these models have not only elucidated mechanisms of resistance but also directly contributed to the discovery and validation of novel therapeutic strategies:(i)PI3K inhibitors and CDK4/6 inhibitors emerged from insights gained through resistant 2D and PDX models [71].(ii)PARP inhibitors were developed through studies in BRCA1/2-deficient GEMMs [68,72].(iii)Immune checkpoint inhibitors and TME-targeted therapies have been evaluated in organoid models and immune-competent GEMMs [67,73].(iv)AI-based in silico models have facilitated target prediction and drug repurposing strategies, although biological validation remains a bottleneck [70].

Thus, experimental models serve a dual purpose: they are essential not only for uncovering mechanisms driving drug resistance but also for screening, optimizing, and validating next-generation therapeutics aimed at overcoming these resistance barriers. Despite their utility, no single model fully recapitulates human BC’s complexity. Each model has inherent limitations in terms of its physiological relevance, scalability, cost, and translational predictability (as summarized in Table 2). Therefore, an integrative, multi-model approach combining in vitro, in vivo, and in silico systems represents the most promising strategy to accelerate the discovery of resistance mechanisms and the development of effective therapies. Taken together, the advancement and optimization of experimental models remain foundational to combating BC drug resistance and reducing the staggering mortality associated with it.

### 4.2. Resistance Due to Genetic Mutations

Mutations in the genes encoding drug targets can directly impede drug binding and reduce efficacy. For instance, mutations in *ERBB2 (HER2)* cause constitutive HER2 activation, conferring resistance to trastuzumab [77,78]. Similarly, specific mutations in the estrogen receptor gene (*ESR1*), such as Y537S, are well-known drivers of endocrine therapy resistance [79]. Beyond direct target alterations, target overexpression (e.g., *HER2*) can overwhelm inhibitors, while target loss (e.g., ER downregulation) renders therapies ineffective. Conversely, the loss of a target, such as the downregulation of ER expression, can render the corresponding targeted therapy ineffective [80]. Enhanced DNA damage response (DDR) pathways also play a critical role in such resistance. These pathways, including homologous recombination repair (HRR), enable BC cells to repair DNA damage induced by chemotherapies (e.g., cisplatin) and PARP inhibitors. In TNBC with *BRCA* mutations, PARP inhibitors target single strand break repair. However, epigenetic modifications, such as *BRCA1* promoter hypermethylation, can initially sensitize cells to PARP inhibition but may subsequently contribute to resistance upon prolonged exposure [81,82]. These findings highlight the significant role of epigenetic regulation in driving resistance.

Circulating tumor DNA (ctDNA) offers a non-invasive, real-time window into tumor evolution, enabling dynamic monitoring of resistance mechanisms. In hormone-receptor-positive (HR^+^) BC, ctDNA frequently detects *ESR1* mutations (e.g., Y537S) [82], which confer endocrine resistance, as well as *PIK3CA* mutations [83,84], which predict resistance to CDK4/6 inhibitors and guide the use of alpelisib. In TNBC, ctDNA reveals *TP53* and *RB1* alterations [85] associated with chemotherapy resistance and may detect acquired *HER2* amplification [45], indicating therapeutic escape. Emerging AI-driven ctDNA analytics [86,87] can predict immunotherapy resistance with an estimated accuracy of ~78%. Collectively, ctDNA profiling facilitates personalized treatment by capturing the molecular dynamics of resistance in a minimally invasive and clinically actionable format [88].

### 4.3. Triple-Negative Breast Cancer (TNBC): Pathobiology and Therapy

TNBC accounts for approximately 11–15% of breast cancers and is defined by the absence of ER, PR, and HER2 expression according to ASCO/CAP standards [83]. These tumors are high grade and genomically unstable, marked by TP53 mutations and BRCA1/2 dysfunction, which drive early relapse and poor prognosis. Five-year survival falls from ~78% in early-stage disease to ~15% in metastatic cases [34].

Molecular heterogeneity is well established: Lehmann’s subtypes (BL1, BL2, immunomodulatory, mesenchymal, mesenchymal stem-like, LAR) and Fudan’s IHC-based groups (LAR, immunomodulatory, basal-like immune-suppressed, mesenchymal-like) guide biological understanding and precision trials. Current therapies rely on chemotherapy—anthracycline/taxane backbones and platinum agents—improving pathologic complete response (pCR) rates and potentially survival in select settings [84,85]. Adjuvant capecitabine benefits patients with residual disease post-neoadjuvant therapy. The PARP inhibitor olaparib prolongs invasive disease-free and overall survival in germline BRCA-mutated patients [86,87]. Immunotherapy has reshaped management: pembrolizumab plus chemo enhances event-free and overall survival in early TNBC (KEYNOTE-522) [85]. In metastatic TNBC, sacituzumab govitecan extends progression-free and overall survival, fulfilling an unmet need in refractory cases (ASCENT trial) [88]. Despite these advances, TNBC remains the most lethal breast cancer subtype, underscoring the urgent need for biomarker-driven and combination therapies.

### 4.4. Resistance Due to Drug Efflux

Chemotherapy, a cornerstone of BC treatment, often encounters resistance mediated by survival pathways and upregulation of efflux pumps [89,90,91]. Overexpression of ATP-binding cassette (ABC) transporters, such as P-glycoprotein (P-gp/ABCB1) and multidrug resistance-associated protein 1 (MRP1/ABCC1), is a major mechanism of multidrug resistance (MDR) in BC [92,93]. Furthermore, BC Resistance Protein (BCRP/*ABCG2*) is another efflux transporter that has been shown to have a significant role in therapy resistance (Figure 2). To overcome drug efflux due to BCRP proteins like ABCG2, inhibitors are being developed [94].

These transporters actively pump drugs, such as doxorubicin and paclitaxel, out of the cell, lowering intracellular concentrations below therapeutic thresholds [92]. In TNBC, the overexpression of ABCB1 is a significant driver of multidrug resistance (MDR). This response is often induced by prior chemotherapy [32,93]. The ATP-dependent efflux process can be further amplified by inflammation, with cytokines like IL-6 and TNF-α upregulating the expression of these transporters [95].

Additionally, alterations in drug metabolism, mediated by changes in the activity of cytochrome P450 enzyme activity, can reduce chemotherapeutic efficacy by affecting drug activity or clearance [96]. The recent literature demonstrates that nanoparticle-based drug delivery systems can bypass these efflux pumps by multiple mechanisms [97], thus enhancing treatment efficacy.

#### Molecular Signaling in Drug Resistance

Drug resistance is governed by multiple cell-signaling mechanisms. Activation of the PI3K/AKT/mTOR cell survival pathway within the TME can promote proliferation, leading to resistance to various therapies, including endocrine therapy and chemotherapy [98]. Similarly, activation of the MAPK pathway can contribute to resistance through multiple mechanisms by enhancing proliferation and survival [99]. The CXCL12/CXCR4 signaling pathway promotes immune suppression, increases fibrosis, and limits infiltration of cytotoxic immune cells in breast tumors [100]. The PI3K/AKT and MAPK pathways mediate resistance by promoting tumor cell proliferation and survival, particularly in response to targeted therapies [98,99]. The TGF-β signaling pathway is involved in immune evasion and enhances the epithelial-to-mesenchymal transition (EMT), thereby increasing metastatic potential [101]. These molecular signaling events are depicted in Figure 2.

### 4.5. Evading Apoptosis in Breast Cancer Resistance

For maintaining tissue homeostasis, regulated cell death (RCD), including apoptosis, ferroptosis, pyroptosis, necroptosis, cuproptosis, and autophagy-dependent cell death, is essential [102,103]. In BC, evasion of RCD allows tumor cells to survive treatments, contributing to therapy resistance and tumor progression [104,105]. BC cells evade apoptosis through several mechanisms. Overexpression of ATP-binding cassette (ABC) transporters (ABCB1, ABCB2, ABCG2), such as P-glycoprotein, expels chemotherapeutic drugs, reducing their efficacy [93]. Mutations in p53 or BCL-2 family proteins disrupt apoptotic signaling, enabling cell survival under stress [106]. Enhanced DNA repair enzymes, like topoisomerase II, or βIII-tubulin overexpression confer resistance to DNA-damaging or microtubule-targeting agents [107,108].

In hormone-receptor-positive BC, *ESR1* mutations promote estrogen-independent tumor growth. On the other hand, activation of the PI3K/Akt/mTOR pathway reduces endocrine therapy effectiveness [109,110]. Additionally, BC cells counteract reactive oxygen species (ROS) to avoid oxidative-stress-induced death [111]. Non-apoptotic RCD evasion, such as resistance to ferroptosis or pyroptosis, further contributes to BC’s adaptability, though specific mechanisms remain under investigation [104].

Targeting alternative RCD pathways, such as pyroptosis, cuproptosis, and ferroptosis, offers promising strategies to overcome BC resistance [112]. For example, Elesclomol, an experimental drug, induces cuproptosis by disrupting mitochondrial metabolism [113]. Sorafenib, combined with ursolic acid, triggers ferroptosis by depleting glutathione and increasing lipid peroxidation [114]. Doxorubicin and tetraarsenic hexaoxide activate pyroptosis via ROS and caspase-3/GSDME pathways, particularly in triple-negative BC [115]. Rapamycin or metformin exploits autophagy to induce cell death [116]. These therapeutic approaches, though varying in clinical development, provide novel avenues to counter BC’s resistance to conventional treatments.

Taken together, evasion of regulated cell death (RCD) has emerged as a defining hallmark of cancer and a central feature of therapeutic resistance. RCD intersects with diverse resistance mechanisms, including genetic and epigenetic alterations, pharmacokinetic adaptations such as drug efflux, and profound metabolic rewiring. These processes ultimately converge on the metabolic–immune axis within the tumor microenvironment, where they recalibrate cell death thresholds and reinforce tumor survival. The consequences extend to angiogenesis, invasion, metastasis, and sustained evasion of immune destruction.

### 4.6. Tumor Microenvironment in Cancer Resistance

The BC-TME is a dynamic and complex ecosystem comprising cancer cells, stromal components, vasculature, and infiltrating immune cells. Increasing evidence underscores the TME’s pivotal role in tumor progression, immune evasion, and resistance to both conventional and targeted therapies [13,117,118]. Across the subtypes—HR^+^, HER2^+^, and TNBC—the TME contributes to resistance through mechanisms such as immunosuppression, fibrosis, and dysregulated stromal–tumor signaling [119]. Key cellular contributors include cancer-associated fibroblasts (CAFs), which deposit dense extracellular matrix (ECM) and secrete pro-survival cytokines (e.g., TGF-β, HGF, IL-6), thereby enhancing drug resistance via PI3K/AKT and MAPK pathways [120]. The regulatory T-cells (Tregs) and myeloid-derived suppressor cells (MDSCs) inhibit cytotoxic T-cell responses (see Figure 3).

In HER2^+^ BC, TAMs suppress antibody-dependent cellular cytotoxicity (ADCC), thereby compromising trastuzumab efficacy [121]. Concurrently, PIK3CA mutations—found in 30–40% of HER2^+^ tumors—sustain PI3K/AKT/mTOR signaling despite HER2 blockade, promoting resistance [122,123]. In TNBC, CCL2-mediated recruitment of immunosuppressive myeloid cells limits immunotherapeutic responses [124,125]. Hypoxic TME conditions—driven by HIF-1α—exacerbate resistance by inducing ECM remodeling, angiogenesis, and direct modulation of drug metabolism [126,127].

#### 4.6.1. Tumor Vasculature in Resistance

Aberrant angiogenesis, driven by vascular endothelial growth factor (VEGF) overexpression, is a hallmark of BC progression and therapeutic resistance [128]. Tumor vasculature exhibits structural and functional abnormalities—leakiness, tortuosity, and poor perfusion—that impair drug delivery, exacerbate hypoxia, and foster an immunosuppressive TME [128,129]. This section examines the role of dysfunctional vasculature in resistance, its interplay with immunosuppression, and emerging strategies to enhance treatment efficacy [129,130].

VEGF-driven angiogenesis in BC creates disorderly, permeable vessels that elevate interstitial fluid pressure and induce severe hypoxia within the TME [128,131,132]. Hypoxia activates hypoxia-inducible factor-1α (HIF-1α), promoting cancer cell survival and resistance to chemotherapy and targeted therapies, such as trastuzumab, particularly in triple-negative BC (TNBC) [128]. The TME subsequently recruits immunosuppressive cells, including regulatory T-cells (Tregs), myeloid-derived suppressor cells (MDSCs)., Additionally, M2-polarized tumor-associated macrophages (TAMs) also inhibit cytotoxic T-cell activity and thus contribute to immune evasion [130]. VEGF further suppresses immunity by inhibiting dendritic cell maturation and expanding immunosuppressive populations, amplifying resistance to immunotherapies, including exploratory CAR-T-cell therapies [133,134]. These concepts were analyzed and explored in detail by Fukumura et al. [130,133].

The seminal work on vascular normalization demonstrates that anti-VEGF therapies, such as bevacizumab, transiently restore vessel function, improving perfusion and drug delivery while reprogramming the TME to enhance immunotherapy outcomes [129,130,135]. By reducing hypoxia and immunosuppressive cell recruitment, normalization sensitizes tumors to immune checkpoint inhibitors and may improve CAR-T-cell infiltration in TNBC, though toxicities remain a challenge [136,137]. Emerging strategies combine vascular normalization with stroma-targeting agents (e.g., FAP or TGF-β inhibitors) and immune checkpoint blockade to dismantle TME barriers [138]. Dr. Jain hypothesized that not only blood vessels but also other components of the TME are abnormal and all these abnormalities in concert fuel tumor progression and treatment resistance [129]. Integrated approaches combining vascular normalization with stroma-targeting agents (e.g., FAP or TGF-β inhibitors) and immune checkpoint blockade, as evidenced by clinical trials (e.g., NCT03394287 for VEGFR2 inhibitor and anti-PD-1 combinations), aim to overcome resistance and enhance outcomes in BC [139].

#### 4.6.2. Breast Cancer Stem Cells in Resistance

BC stem cells (BCSCs) are a rare tumor subpopulation that contribute to tumorigenic capacity and therapeutic resistance through self-renewal and plasticity. IL-6, hypoxia, VEGF and other cell signaling molecules collectively promote BCSC plasticity and long-term survival [140,141,142]. These traits allow them to initiate, sustain, and regenerate tumors, driving heterogeneity, metastasis, and relapse [143,144]. Unlike bulk tumor cells, BC stem cells (BCSCs) are highly resilient, persisting after therapy and replenishing the tumor mass, which drives treatment failure. Their resistance arises from mechanisms such as enhanced DNA repair, drug efflux through transporters like ABCB1 and ABCG2, quiescence, and activation of survival signaling pathways, including PI3K/AKT and Bcl-2. These traits are regulated by developmental pathways—Notch, Wnt/β-catenin, Hedgehog, and Hippo—that sustain BCSC stemness and survival [145,146,147]. Agents like evofosfamide and salinomycin have shown preclinical efficacy in targeting hypoxia-adapted and stem-like cells, respectively [148,149]. Additional strategies targeting Notch signaling (e.g., ATRA, γ-secretase inhibitors) reduce the stem-like subpopulation [150]. Despite promising preclinical data, clinical translation remains challenging. Rational combinations of BCSC-targeted agents with standard therapies, supported by robust biomarkers to track BCSC dynamics, are needed to ensure durable responses [141,151,152].

#### 4.6.3. BC Stem Cell Dormancy

Each dormancy state is associated with unique metabolic adaptations—including OXPHOS and ROS scavenging. High ALDH activity, particularly ALDH1, further aids in drug detoxification [153]. This includes cytokines (e.g., IL-6, CXCL8), growth factors, hypoxia, and stromal cells like CAFs, which maintain BCSC stemness and promote immune evasion [147,154]. EMT, often induced by TME signals, triggers stem-like traits and enhances BCSC migratory capacity, directly contributing to metastasis [140]. These properties are reinforced by cues from the TME. The TME autophagy and unfolded protein response (UPR) enable tumor cells to evade therapy and immune surveillance. Reactivation of these dormant cells may lead to tumor relapses or secondary metastases. BCSCs can invade tissues, intravasate, survive circulation, and colonize distant sites, underscoring their role in the metastatic cascade. These cells also evade immune attack by expressing PD-L1, secreting immunosuppressive cytokines (e.g., TGF-β, IL-10), and recruiting Tregs and MDSCs, shaping an immunosuppressive niche [155,156].

Dormancy is a critical BCSC feature that allows them to escape therapy and persist as minimal residual disease [155]. Dormant BCSCs exist in two main forms: (1) cellular dormancy (quiescent G0-phase cells) and (2) angiogenic dormancy (small clusters without sufficient vasculature). These cells may lie latent for years and reactivate under favorable TME cues, triggering relapse [157].

Targeting BCSCs is essential for durable BC control [158]. Approaches include (a) pathway inhibition: small molecules or antibodies targeting Notch (e.g., MK-0752), Wnt (e.g., LGK974), and Hedgehog signaling are in early-phase trials for reducing BCSC populations [156,159]; (b) immunotherapy: preclinical efforts using CAR-T-cells directed at BCSC-specific antigens show promise [141]; and (c) differentiation therapy: agents like all-trans retinoic acid (ATRA) may induce BCSC differentiation, sensitizing them to chemotherapy [159].

#### 4.6.4. Role of Tumor-Associated Macrophages in Immunosuppression

T-cell exhaustion is driven by two main factors within the TME: persistent antigen exposure and immunosuppressive cues from TAMs. This includes upregulation of inhibitory receptors (PD-1, CTLA-4, TIGIT) and suppression of cytokine production (IFN-γ, IL-2, TNF) in CD8^+^ T-cells [134,160,161]. The exhausted T-cells limit the efficacy of immunotherapies [162]. Therefore, TAM-mediated T-cell exhaustion is a therapeutic focus. Engineering exhaustion-resistant CD8^+^ T-cells (e.g., via TOX knockdown) and combining ICB with TME modulators represent synergistic strategies [163]. Cold TMEs are immunologically inert, featuring stromal exclusion, hypoxia, and limited immune infiltration. In contrast, hot TMEs exhibit T-cell infiltration, normalized vasculature, and enhanced immune responsiveness (Figure 3). Emerging omics-driven technologies and rationally designed therapies aim to convert cold tumors into hot, immuno-responsive states, offering new avenues to enhance clinical efficacy in BC.

#### 4.6.5. Role of Exosomes in TME Modulation

Exosomes play a pivotal role in propagating drug resistance by transferring key molecular cargo such as miRNAs, lncRNAs, and proteins that reprogram recipient cells toward epithelial–mesenchymal transition (EMT), stemness, and drug efflux phenotypes [46,164]. Hypoxic and acidic TME conditions also upregulate exosome biogenesis (e.g., via HIF-1α signaling), increasing release of immunosuppressive exosomes from tumor cells. TME components like cytokines (e.g., TGF-β) and stromal cells enhance exosome uptake by immune cells via receptor-mediated endocytosis or membrane fusion [46,164,165,166]. Exosomes carry inhibitory molecules (e.g., PD-L1, miRNAs like miR-21) that suppress T-cell activation, induce regulatory T-cells, and promote M2 macrophage polarization, dampening antitumor immunity [47,166].

In BC, tumor-derived exosomes contribute to doxorubicin resistance by stabilizing HIF-1α and delivering lncRNA H19 to adjacent cells, thereby reinforcing a resistant microenvironment [165,167]. Notably, silencing HIF-1α diminishes exosomal H19 levels and restores sensitivity to chemotherapy. Similarly, exosomes from CAFs enhance tumor stemness and therapy resistance by transferring regulatory miRNAs and signaling proteins.

### 4.7. Breast Cancer Metastasis to Bone

Like other cancers, the progression of breast cancer leads to the establishment of secondary lesions away from the site of origin [168]. This involves multiple organs throughout the body, including bone, lung, liver and brain. Bone is the most common site of distant metastasis in advanced breast cancer, occurring in up to 70% of patients with metastatic disease [169,170]. Approximately half of patients who experience a relapse will have the bone as their first site of metastasis. Approximately 70–85% of patients with advanced-stage disease have been shown to be affected by an altered bone microenvironment [170]. Metastasis of the skeleton leads to debilitating symptoms, including pathological fractures, spinal cord compression, pain, and hypercalcemia, significantly reducing patient survival and quality of life [169].

The skeletal microenvironment containing fenestrated capillaries in the bone matrix favors metastatic colonization through its rich vascular supply, remodeling activity, and abundant growth factors [169,170]. Once tumor cells arrive, they initiate a pathological interaction commonly referred to as the ‘vicious cycle’ [171,172]. Tumor-secreted factors such as PTH-related protein (PTHrP), interleukin-6 (IL-6), IL-11, and prostaglandin E_2_ stimulate osteoblasts and stromal cells to increase the expression of RANKL [170,173]. This promotes osteoclast differentiation and bone resorption that releases matrix-bound growth factors such as transforming growth factor-β (TGF-β) and IGFs [173]. These factors, in turn, enhance tumor proliferation and further PTHrP production, reinforcing the cycle [169,171].

MicroRNAs modulate key steps in this metastatic cascade. miR-10b, for instance, is upregulated in metastatic breast cancer and promotes invasion and early bone colonization [174,175]. Therapeutic strategies currently focus on disrupting the vicious cycle and preserving skeletal integrity. Bisphosphonates, such as zoledronic acid, inhibit osteoclast-mediated bone resorption by disrupting essential intracellular functions within osteoclasts [176]. Denosumab, a monoclonal antibody against RANKL, has been shown to reduce skeletal-related events and delay pain progression more effectively than bisphosphonates in multiple randomized trials [177,178]. Early detection through advanced imaging modalities (e.g., PET/CT, whole-body MRI) and biomarker surveillance, including circulating tumor DNA and bone turnover markers, may allow preemptive intervention, potentially delaying or preventing the onset of skeletal complications [179,180,181].

### 4.8. The TME as a Major Hub of Resistance

The survival of BC cells is sustained not only in isolation but also through a network of stromal and immune cells, oncoproteins, non-coding RNAs, microbes, and exosomes that collectively reinforce resistance [11,14,17]. Within this network, the TME and cellular metabolism function as a central hub where genetic and epigenetic alterations, signaling pathways, and extracellular cues converge [117,182]. These influences recalibrate thresholds for RCD, preserve cancer stem cell plasticity, and enable angiogenesis, invasion, and metastasis [128,129]. Microbial communities and exosome-mediated communication add further adaptability by shaping immune evasion and remodeling the surrounding stroma. The metabolic and immune axes are equally decisive, as nutrient competition, lactate accumulation, and mitochondrial dysfunction suppress effective immunity while enhancing drug efflux programs [13]. Resistance, therefore, emerges as a hub that sustains proliferation and long-term persistence of BC. Advances in nanotechnology provide opportunities to disrupt this hub [183]. Multifunctional nanocarriers can bypass efflux barriers, remodel the tumor milieu, and enhance therapeutic delivery [184,185]. Future therapies will be shaped by precise insight into how the metabolic and immune axis integrates genomic, microbial, and stromal drivers to set cell death thresholds and determine the response to immunotherapy.

## 5. Metabolic Reprogramming in BC Resistance

Metabolic homeostasis is a cornerstone of normal physiological function at both the organismal and cellular levels. In BC, this homeostatic balance is profoundly disrupted within the tumor microenvironment, driving malignant progression, therapeutic resistance, and immune evasion [186,187,188]. Tumor cells, stromal components, and immune effectors undergo subtype-specific metabolic reprogramming—a hallmark of cancer—to sustain proliferation, adapt to therapeutic stress, and subvert antitumor immunity [188]. In BC, tumor cells rewire glycolysis, glutaminolysis, fatty acid oxidation, and mitochondrial dynamics, while stromal cells enhance lactate production and immune cells face nutrient deprivation, collectively fostering a metabolically hostile TME [9,186,187,189]. Hypoxia, nutrient scarcity, and TME acidosis impose selective pressures that promote metabolic plasticity, enabling resistance to chemotherapy, targeted therapies, and immunotherapies across hormone-receptor-positive (HR^+^), HER2-positive (HER2^+^), and TNBC subtypes. This section elucidates how these metabolic shifts, intricately linked to immune suppression and therapeutic failure, pose formidable barriers to effective BC treatment.

### 5.1. Aerobic Glycolysis vs. Oxidative Phosphorylation

Otto Warburg proposed that cancer cells prefer glycolysis over oxidative phosphorylation (OXPHOS) due to dysfunctional mitochondria [190]. While groundbreaking, this hypothesis is now recognized as incomplete. Later studies, including Warburg’s own, did not consistently demonstrate defective mitochondrial respiration as a hallmark of malignancy [191]. In fact, mitochondrial respiration and related functions are now understood to be critical for tumor progression and immune evasion [187]. In BC, the preference for aerobic glycolysis is largely driven by oncogenic signaling pathways such as PI3K/AKT, MYC, and HIF-1α. These pathways upregulate key glycolytic enzymes, including hexokinase 2, pyruvate kinase M2, and lactate dehydrogenase A (LDHA), supporting anabolic processes such as nucleotide and lipid synthesis rather than merely producing ATP [187,192]. Importantly, most BC cells retain intact mitochondria and the capacity for OXPHOS. This metabolic flexibility—often termed metabolic plasticity—allows cancer cells to switch between energy sources depending on environmental pressures. Imaging studies using fluorodeoxyglucose positron emission tomography (FDG-PET) have linked high glucose uptake not only to rapid proliferation but also to dynamic remodeling of the tumor microenvironment [187].

Role of Lactate in Cancer Metabolism

Lactate, a byproduct of cancer cell metabolism, is exported from tumor cells via monocarboxylate transporters (MCT1 and MCT4), leading to acidification of the TME to a pH of 6.5–6.8. This acidic environment impairs the function of cytotoxic T lymphocytes (CTLs) and natural killer (NK) cells, suppresses interferon-γ production, and promotes an ‘immune-cold’ TME. These effects are especially pronounced in hormone-receptor-positive (HR^+^) and HER2-positive BCs, which often show poor responses to immune checkpoint inhibitors [193,194]. In TNBC, cancer cells rely heavily on glycolysis, which is linked to a pathway that helps them neutralize harmful reactive oxygen species (ROS) produced by chemotherapy drugs like anthracyclines [187]. Concurrently, IL-6/HIF-1α signaling amplifies glycolytic flux and lactate export, compounding extracellular acidification and further hindering drug diffusion [195].

Immunosuppression by Aerobic Glycolysis in BC

This dual function of aerobic glycolysis—fueling biosynthetic pathways while shaping an immunosuppressive TME—underscores its central role in breast cancer’s therapeutic resistance. Nutrient depletion within the TME impairs the function of cytotoxic T lymphocytes and natural killer cells by limiting the energy required for their proliferation and tumor-killing activity [196]. Accumulated lactate further disrupts T-cell receptor signaling and glycolytic metabolism, while IL-6-mediated stabilization of HIF-1α drives excessive glycolysis. Together, these changes create a self-reinforcing loop that promotes immunosuppression and therapeutic resistance. In TNBC, this ‘metabolic–immune axis’ not only shields tumors from immune surveillance but also reduces drug penetration by altering the TME pH and upregulating angiogenic factors like VEGF [197]. This dual role—immune evasion and chemoresistance—positions metabolic rewiring as a linchpin of BC’s adaptive resilience, necessitating integrated therapeutic strategies to disrupt this vicious cycle [193,195].

As the glycolytic capacity saturates BC cells, particularly HR^+^ subtypes, they exhibit glutamine addiction to sustain mitochondrial bioenergetics under therapeutic stress [198]. Glutaminase 1 (GLS1) converts glutamine into glutamate, replenishing the tricarboxylic acid (TCA) cycle for anaplerosis and glutathione synthesis to counter oxidative stress [199,200]. In HR^+^ BC treated with aromatase inhibitors, GLS1 and ASCT2 upregulation maintains redox homeostasis and biomass production. In TNBC, glutaminolysis supports rapid proliferation and ROS detoxification, enabling survival during chemotherapy [201]. On the other hand, glutamine starvation of T-cells significantly hinders T-cell proliferation and cytokine production. Collectively, metabolic plasticity—toggling between glycolysis, glutaminolysis, and fatty acid oxidation—allows BC cells to adapt to nutrient scarcity and therapeutic pressures, rendering single-pathway targeting ineffective [202].

### 5.2. Role of Mitochondria in BC Resistance

Mitochondria have emerged as key modulators of therapeutic resistance in BC, influencing not only cellular metabolism but also apoptosis, redox balance, and immune responses [203,204]. Beyond their canonical role as ATP generators, mitochondria act as integrative hubs for signaling pathways that support tumor progression and adaptation to treatment. Mechanistically, mitochondrial dysfunction and plasticity contribute to BC resistance primarily through mitochondrial DNA (mtDNA) mutations, metabolic heterogeneity, dysregulated mitochondrial dynamics, and intercellular mitochondrial transfer [203,205]. This section critically examines how mitochondrial genetics, functional variability, and organelle exchange contribute to therapeutic failure across BC subtypes. These mechanisms not only fuel metabolic adaptability but also compromise antitumor immunity, particularly through mitochondrial hijacking of T-cells and other immune effectors [205,206].

The mitochondrial genome encodes 13 essential proteins of the electron transport chain (ETC), yet it remains highly susceptible to damage due to its proximity to ROS, lack of histone protection, and limited DNA repair capacity [207]. In BC, recurrent somatic mutations in genes such as MT-ND1, MT-ND4, and MT-ND5 (Complex I subunits) have been documented [208]. Rather than impairing respiration, these mutations often promote metabolic plasticity, enabling cancer cells to tolerate oxidative stress or shift toward an OXPHOS-favorable state under treatment pressure. Some mtDNA variants are associated with aggressive phenotypes, altered redox signaling, and resistance to genotoxic therapies [192,209,210]. Additionally, mtDNA mutations may influence immune recognition by altering mitochondrial antigen presentation. Altogether, mtDNA instability contributes to BC progression and its signatures may serve as both biomarkers and therapeutic targets.

‘Mitochondrial heterogeneity’ represents variations in organelle mass, membrane potential, ROS production, and metabolic behavior, which is a defining feature of treatment for resistant breast tumors. HR^+^ subtypes often rely on OXPHOS and fatty acid oxidation, whereas TNBCs display heightened glycolytic and glutamine metabolism [211]. Even within a single tumor, diverse mitochondrial phenotypes coexist, allowing subpopulations of cells to escape metabolic or drug-induced stress [209,212]. A well-characterized resistance mechanism involves dysregulated mitochondrial dynamics. Dynamin-related protein 1 (Drp1)-mediated mitochondrial fission leads to fragmented mitochondria with reduced ROS production, attenuated pro-apoptotic signaling, and increased resistance to stress. In HER2-positive tumors, elevated Drp1 activity has been linked to trastuzumab resistance [213].

Additionally, overexpression of anti-apoptotic Bcl-2 family proteins—including Bcl-2, Mcl-1, and Bcl-xL—prevents cytochrome *c* release by inhibiting mitochondrial outer membrane permeabilization (MOMP), thereby inhibiting intrinsic apoptotic pathways [214]. In TNBC, PGC-1α-driven mitochondrial biogenesis augments OXPHOS capacity, facilitating survival under chemotherapy-induced metabolic stress [215]. Somatic mutations in MT-ND4 and other ETC genes further enhance ETC efficiency, promoting metastasis and potentially altering immunogenicity [216]. These mitochondrial adaptations form a multifaceted resistance system supporting energy production, limiting apoptosis, and facilitating immune escape.

#### Intercellular Mitochondrial Transfer

One of the most remarkable features of tumor mitochondrial biology is their ability to traffic between cells. Intercellular mitochondrial transfer occurs via tunneling nanotubes (TNTs), microvesicles, gap junctions, and cell fusion, forming a dynamic exchange network between tumor cells and stromal or immune cells [217,218]. Mitochondrial dysfunction in T-cells can cause T-cell exhaustion via defective mitochondrial transfer through tunneling nanotubes [205]. This can further weaken antitumor immunity. While the implications of this mitochondrial ‘networking’ are still being defined, it clearly contributes to bioenergetic resilience and cellular reprogramming under therapy-induced pressure. As shown in Figure 4, mitochondria can play a multifaceted role of in BC resistance.

Key resistance mechanisms include (1) mtDNA mutations in Complex I (e.g., MT-ND4) that promote ETC efficiency and metastasis; (2) mitochondrial heterogeneity and dynamic remodeling that enable metabolic plasticity and apoptotic resistance; and (3) intercellular mitochondrial transfer through tunneling nanotubes (TNTs), vesicles, and fusion events. An emerging resistance axis involves mitochondrial hijacking, where BC cells transfer depolarized or dysfunctional mitochondria into CD4^+^ and CD8^+^ T lymphocytes, particularly within the TME [205]. These mitochondrial adaptations contribute to treatment resistance and immune evasion.

Immune suppression in BC may occur independent of classical immune checkpoint pathways, potentially explaining the poor response to checkpoint blockade in some patients with apparent T-cell infiltration [219]. Recent studies suggest that disruption of TNTs [220,221] using cytoskeletal inhibitors such as cytochalasin B or latrunculin A, or inhibition of mitochondrial trafficking proteins like Miro1, has shown potential in preclinical models to restore immune competence [218,222].

However, the physiological roles of TNTs in normal tissues raise safety concerns for systemic targeting. Mitochondrial transfer thus plays a dual role in the TME: (1) conferring metabolic flexibility to cancer cells, and (2) attenuating immune effector function through intercellular energy redistribution—highlighting the immuno-metabolic complexity of the TME. Together, mitochondria in BC serve roles beyond ATP generation; they actively contribute to therapy resistance, immune escape, and cellular plasticity. Their heteroplasmic variation, dynamic morphological behavior, and capacity for intercellular trafficking complicate efforts to develop durable therapies. Future therapeutic strategies should focus on disrupting key mitochondrial resistance mechanisms. These include inhibiting mitochondrial fission and fusion dynamics (e.g., Drp1 inhibitors such as Mdivi-1), blocking TNT-mediated organelle trafficking, targeting anti-apoptotic mitochondrial pathways, and designing biomarker-guided combinations of metabolic and immunotherapeutic agents. Emerging platforms such as live-cell mitochondrial imaging, spatial metabolomics, and single-cell transcriptomic profiling will be instrumental in identifying mitochondrial phenotypes in clinical samples. These tools may enable a precision oncology framework that exploits mitochondrial vulnerabilities as next-generation therapeutic targets.

### 5.3. Therapeutic Targeting of BC Metabolism

Targeting the metabolic vulnerabilities of BC offers a promising strategy to overcome therapeutic resistance. However, translating preclinical successes to the clinic remains challenging. Inhibitors like 2-deoxyglucose (2-DG) disrupt glycolysis by blocking hexokinase, thus enhancing the efficacy of chemotherapy in BC models by starving tumors of energy and biosynthetic precursors [192]. Similarly, CB-839 (telaglenastat), a glutaminase 1 (GLS1) inhibitor, synergizes mTOR inhibitors and DNA-damaging agents in TNBC, reducing tumor growth by approximately 40% in xenograft models [201]. Dichloroacetic acid (DCA) shifts metabolism from glycolysis to oxidative phosphorylation (OXPHOS) by inhibiting pyruvate dehydrogenase kinase. This drastically promotes apoptosis in BC cells when combined with photodynamic therapy [186].

Despite these advances, clinical trials, such as a phase II study of CB-839 in TNBC (NCT02771626), have shown modest efficacy, hampered by systemic toxicity and the metabolic heterogeneity of BC subtypes [223]. Furthermore, metabolic epigenetic interactions complicate targeting efforts, as metabolites like acetyl-CoA fuel histone acetylation, driving adaptive gene expression that sustains resistance [224]. Combined approaches, integrating metabolic inhibitors with targeted therapies or epigenetic modulators, are more effective at overcoming these barriers and restoring therapeutic sensitivity.

Thus, metabolic reprogramming is one of the primary drivers of BC resistance and immune evasion, demanding integration into precision oncology. Combining metabolic inhibitors (e.g., CB-839, mdivi-1) with targeted therapies or ICIs could disrupt the immunosuppressive TME. Biomarkers like GLS1 expression, 18F-fluoroglutamine PET, or lactate levels are critical for patient stratification [223]. Targeting metabolic–epigenetic interactions and TNT-mediated immune sabotage offer novel avenues to overcome resistance, particularly in immunologically refractory TNBC.

Targeting TME heterogeneity remains a major challenge. Lifestyle interventions—such as diet, exercise, and stress reduction—may positively modulate the TME composition [225]. Clinically, PD-1/PD-L1 inhibitors (e.g., pembrolizumab) have shown efficacy in TNBC, with the KEYNOTE-522 trial demonstrating improved event-free survival in early-stage disease [73,85]. Other promising avenues include (a) IL-15 agonists to activate NK cells [226]; (b) CXCL12/CXCR4 axis blockade to enhance immune infiltration [227]; (c) TGF-β inhibition, which restores T-cell function and blocks EMT [228,229,230]; and (d) indoleamine 2,3-dioxygenase 1 (IDO1), a tryptophan-catabolizing enzyme, enhances immunosuppression in metastatic BC, with its inhibition reducing TNBC invasiveness [231,232]. CSF1R blockade reprograms tumor-associated macrophages (TAMs) from protumor (M2-like) to antitumor (M1) phenotypes, enhancing T-cell infiltration in TNBC [233,234]. Inhibiting fibroblast activation protein (FAP) or TGF-β signaling reduces immune exclusion by cancer-associated fibroblasts (CAFs) [233,235]. Blocking IL-6, TGF-β, or the CXCL12/CXCR4 axis further enhances T-cell trafficking and synergizes with ICIs [182].

Combination regimens involving immunotherapy, chemotherapy, and epigenetic modulators are under active investigation. Liquid biopsy tools offer non-invasive means to monitor resistance mechanisms and tailor therapy in real time. Together, these strategies provide a comprehensive framework to overcome immune evasion and resistance driven by the TME.

### 5.4. Metabolic–Immune Axis

Moving forward, metabolic rewiring in BC extends beyond intrinsic adaptations of tumor cells, emerging as a central axis that links energy production with immune regulation and therapeutic resistance [188,189]. BC cells exhibit flexible fuel utilization and dynamic shifts between glycolysis and oxidative phosphorylation to sustain chronic proliferation and generate biosynthetic precursors [187,192]. This plasticity is further shaped by extrinsic modulators within the TME, including hypoxia, cytokine signaling, stromal interactions, microbial metabolites, and exosomes [1,13]. Collectively, these influences define a metabolic–immune axis that fuels growth, enforces immune suppression, and supports drug resistance and metastatic progression. Mounting evidence suggests that this axis functions as a central determinant of BC pathogenesis and a critical target for future therapeutic intervention.

## 6. Immunotherapy in Breast Cancer

The immune system plays a critical role in surveilling and eliminating BC cells through cancer immunoediting, a process involving antigen release, T-cell priming, and tumor elimination [236,237]. However, the TME in BC disrupts this cycle, promoting immune evasion, thus causing therapeutic resistance. Furthermore, the BC TME establishes a supportive niche where cancer cells interact with immune cells and neighboring endothelial cells and is thus a feasible target for cancer therapy [231]. TNBC shows partial responsiveness to immunotherapy, while hormone-receptor-positive (HR^+^) and HER2-positive (HER2^+^) tumors often remain immune cold due to TME-mediated suppression [238]. Driven by immune checkpoints (PD-1/PD-L1, CTLA-4), pro-tumoral cells (TAMs, CAFs), cytokines (TGF-β, IL-6), and immuno-metabolic alterations, the TME creates an immunosuppressive niche. This section explores immune evasion mechanisms, current and emerging immunotherapies, and strategies to overcome TME barriers, emphasizing integrative approaches to restore immune surveillance and address resistance across BC subtypes.

The PD-1/PD-L1 axis suppresses T-cell activity in the BC TME, promoting immune evasion [239,240]. Immune checkpoint inhibitors (ICIs), such as pembrolizumab and atezolizumab, enhance T-cell responses (KEYNOTE-522 trial) to TNBC chemotherapy by 13% [24,56]. However, resistance persists due to MHC class I downregulation, alternative checkpoints (e.g., LAG-3, TIM-3), and TME immunosuppression [49,241]. CTLA-4 inhibitors, less effective in HR^+^ tumors, show potential in combination with PD-1 inhibitors to enhance T-cell infiltration [238]. Combining ICIs with chemotherapy induces immunogenic cell death, improving antigen presentation, though TGF-β-mediated immunosuppression, which limits broader efficacy [242,243].

### Emerging Immunotherapies

Personalized neoantigen vaccines targeting tumor-specific mutations show immunogenicity in TNBC, using antigens like HER2 and MUC1, though low tumor immunogenicity and TME tolerance limit efficacy [244]. Novel vaccine formulations and delivery strategies are under investigation to address scalability and timing challenges [244]. Chimeric antigen receptor T-cell (CAR-T) therapy, transformative in hematologic malignancies, is exploratory in BC due to the immunosuppressive TME, antigen heterogeneity, and significant toxicities, including cytokine release syndrome (CRS), neurotoxicity (ICANS), and off-target effects causing inflammation from cellular debris [245,246].

In TNBC, CAR-T-cells targeting MUC1 (NCT04025216) and mesothelin (MSLN; NCT02792114) show preclinical promise, with MUC1 overexpressed in ~90% of BCs and MSLN effective in chemoresistant models [247,248,249]. HER2-targeted CAR-T-cells, including bispecific HER2/MUC1 CARs, reduce antigen escape in HER2^+^ BC (NCT04660929) [250]. Multi-armored CAR-T-cells with PD-1 or TGFBR2 knockouts enhance T-cell activity in TNBC models [251]. Emerging CAR-based therapies, such as CAR-macrophages (CAR-M) and CAR-NK cells, show potential in navigating the TME [251]. Combination therapies (e.g., CAR-T with anti-PD-L1 or CDK7 inhibitors) improve efficacy but face challenges from toxicity and high costs [252,253]. These therapies, while promising, require overcoming TME barriers, antigen escape, and manufacturing hurdles to become viable BC treatments.

## 7. Exosomes in Breast Cancer Resistance and Therapy

Both normal and cancer cells secrete exosomes for intercellular communication; however, BC cells secrete significantly more exosomes, positioning them as key mediators of therapy resistance. These vesicles deliver diverse bioactive cargo, including mRNAs, miRNAs, long non-coding RNAs (lncRNAs), proteins, and lipids that modulate the TME to support tumor survival and therapy evasion [46]. Exosome-mediated signaling enhances angiogenesis, immune suppression, and extracellular matrix (ECM) remodeling, contributing to systemic therapy failure [164,165]. Cancer-associated fibroblasts (CAFs), abundant in BC stroma, facilitate tumor progression and resistance through secretion of growth factors, tumor-promoting exosomes, ECM remodeling, and immunosuppression [254].

In TNBC, exosomes transfer resistance-conferring molecules, making TNBC chemo-resistant. For example, ABCB1 mRNA-enriched exosomes enhance P-glycoprotein expression in recipient cells, reducing cisplatin and doxorubicin efficacy [255]. Similarly, exosomal-miR-21 promotes resistance by suppressing PTEN and activating PI3K/AKT signaling [255,256]. Beyond chemoresistance, exosomal lncRNA SNHG16 upregulates PD-L1, attenuating T-cell responses to anti-PD-1 therapies [166]. In hormone-receptor-positive (HR^+^) BC, miR-221 and miR-222 delivered via exosomes downregulate the estrogen receptor (ER), promoting endocrine resistance [255,256]. Pro-inflammatory cytokines such as IL-6 and TNF-α further amplify exosome release and oncogenic signaling [257]. Additionally, exosomal TGF-β promotes epithelial-to-mesenchymal transition (EMT), enhancing metastatic potential [258,259].

CAF-derived exosomes enriched in IL-8 sustain stem-like phenotypes in TNBC cells, reinforcing resistance [260]. CAF-derived exosomes also reprogram tumor metabolism by transferring detoxification enzymes and metabolic intermediates, enhancing survival under therapeutic stress [261]. These insights position CAFs and their exosomal signaling as therapeutic targets, with emerging strategies aimed at disrupting exosome release or blocking downstream signaling pathways [261,262].

Targeting exosome biogenesis is an emerging approach to counter exosome-mediated resistance. Rab27a inhibition, for instance, limits exosome secretion and sensitizes BC cells to chemotherapy [263]. Engineered exosomes carrying miR-134 suppress chemoresistance in TNBC by downregulating STAT5B and Hsp90, enhancing cisplatin sensitivity [264]. Similarly, exosomes from drug-sensitive BC cells delivering miR-765 show potential in re-sensitizing resistant cells [265]. Combining exosome-targeted strategies with TME-modulating agents, such as TGF-β inhibitors or CSF1R antagonists, may offer synergistic benefits [266]. However, clinical translation remains challenging due to exosome production, scalability and tumor heterogeneity.

### Role of MinPP1 in Carcinogenesis

MinPP1 (Multiple Inositol Polyphosphatase) is an inositol-polyphosphate-metabolizing enzyme. It is causally linked to apoptosis in several cancer cells [267]. Our laboratory has recently identified a MinPP1 secreted in BC exosomes and more so under ER stress conditions [268]. Exosomes were traditionally known to originate from multi-vesicular bodies (MVBs) in various mammalian cells. However, emerging evidence implicates the involvement of endoplasmic reticulum (ER) in exosome biogenesis, particularly under ER stress [269,270]. ER-derived proteins detected within exosomes suggest additional roles in cancer progression and therapy resistance. Interestingly, both *MinPP1* and the tumor suppressor gene *PTEN* are in proximity on chromosome 10q23. Both gene products display inositol-phosphate-metabolizing activity and are linked to PI3K/AKT signaling. Loss of this locus, encompassing both *MinPP1* and *PTEN*, is observed in some cancers [271]. Inositol hexakisphosphate (InsP**_6_**), a key substrate of MinPP1, is known to inhibit cell proliferation and induce apoptosis [272,273,274]. Our work demonstrates that MinPP1 secreted in exosomes hydrolyzes InsPs and modulates apoptosis in BC cells [267]. Thus, exosomal secretion of an isoform of MinPP1 represents a novel mechanism of TME modulation by metabolic reprogramming supporting tumor progression. Thus, targeting exosomal MinPP1 could restore InsP**_6_**-mediated tumor suppression. Ongoing research in our laboratory focuses on developing MinPP1-specific inhibitors as potential therapeutic agents against BC.

## 8. Role of Microbiota in Therapeutic Resistance

BC tissues harbor a distinct and dysbiotic microbiome compared to normal breast tissue, significantly influencing tumor biology, immune modulation, and therapeutic outcomes [139,140]. While normal breast tissue contains a conserved microbial community (Proteobacteria, Firmicutes, Actinobacteria, and Bacteroidetes), BC tissues show reduced α-diversity and enrichment of *Escherichia-Shigella*, *Staphylococcus*, and *Fusobacterium* [141,142]. Specifically, *Methylobacterium radiotolerans* and *Sphingomonas yanoikuyae* are elevated in BC tissues [143], with the former potentially supporting tumor survival by modulating oxidative stress and lipid metabolism, while the latter’s depletion is associated with dysbiosis and cancer progression (Figure 5) [142,146]. These microbial shifts appear subtype-specific, with TNBC and estrogen-receptor-positive (ER+) tumors displaying distinct microbial signatures that may influence immune evasion and therapy resistance [144]. In this regard, *Fusobacterium nucleatum* promotes BC cell migration, metastasis, and immune suppression via the miR-21-3p/FOXO3 axis [145]. *Stenotrophomonas maltophilia* has also been linked to reduced CD8^+^ T-cell infiltration, further contributing to immune suppression within the TME [275,276].

Importantly, intra-tumoral bacteria are not limited to extracellular niches but are also localized intracellularly within epithelial cells, fibroblasts, and immune cells (tumor-associated macrophages (TAMs), dendritic cells (DCs), and neutrophils). The intracellular microbes interact with innate immune sensors such as Toll-like receptors (TLRs) and NOD-like receptors, inducing low-grade inflammation, facilitating immune evasion, and promoting metastatic dissemination [143]. These interactions further impair cytotoxic CD8^+^ T-cell infiltration and drive resistance to chemotherapy, endocrine therapy, and immune checkpoint blockade (Figure 5) [143,144]. Angiogenesis, a hallmark of BC progression, is also shaped by tumor-resident and gut microbiota through inflammatory and metabolic signaling pathways [147]. These findings establish that the breast tumor microbiota modulate the TME, influencing immune tone, vascular function, metastasis, and therapeutic resistance [140,148,149,150].

## 9. Nanotechnology in Breast Cancer

Nanotechnology offers innovative solutions to overcome therapeutic resistance in BC by addressing the limitations of conventional chemotherapy, such as non-specific biodistribution, off-target toxicity, and poor tumor penetration [277,278]. Leveraging the TME leaky vasculature via the ‘enhanced permeability and retention (EPR)’ effect, nanotechnology enhances drug delivery to resistant tumors [183,184,185]. This section explores the mechanisms, the clinical applications, and the role of nanoparticles in overcoming resistance [114].

Nanoparticles (NPs) employ passive and active targeting to improve drug specificity. ‘Passive targeting’ exploits the EPR effect, where abnormal tumor vasculature, characterized by leakiness and poor perfusion, allows NP accumulation in the TME [279]. ‘Active targeting’ enhances selectivity by functionalizing NPs with ligands (e.g., antibodies, peptides, aptamers) that bind tumor-specific receptors, such as HER2 or folate receptors in BC [279]. Nanoparticle platforms include liposomes, polymeric NPs, silica NPs, and gold NPs (AuNPs), with AuNPs offering photothermal ablation and tunable surface chemistry but facing translational challenges like dose-dependent toxicity and uncertain long-term clearance [280].

### Use of Nanocarriers for Overcoming Resistance

FDA-approved nanocarriers, such as Doxil^®^ (PEGylated liposomal doxorubicin) and Abraxane^®^ (albumin-bound paclitaxel), improve pharmacokinetics, reduce systemic toxicity, and enhance tumor accumulation in BC [281,282,283,284]. These formulations leverage the EPR effect and benefit from vascular normalization strategies that improve perfusion, enhancing drug delivery to resistant tumors [283]. Thus, nanocarriers can help overcome drug resistance in tumors by increasing the amount of drug that reaches the tumor, making them useful for treating both initial and difficult-to-treat breast cancers [281,283].

BC resistance, driven by efflux pump overexpression (e.g., P-glycoprotein), altered drug metabolism, and defective apoptosis, limits therapeutic efficacy [184]. Nanocarriers bypass efflux transporters and enhance intracellular drug delivery through modified release profiles [279,280]. Co-delivery nanocarriers, combining chemotherapeutics with resistance-modulating agents (e.g., siRNAs targeting MDR1 or PI3K inhibitors), target multiple resistance pathways, improving outcomes in resistant BC [285]. Stimuli-responsive NPs, triggered by tumor-specific cues (e.g., acidic pH, enzymatic activity), offer precise drug release [282]. Emerging stimuli-responsive ‘smart’ nanocarriers offer further precision, with drug release triggered by tumor-specific cues such as acidic pH, enzymatic activity, or oxidative conditions within the TME [286].

Theranostic NPs (multifunctional nanosystems) integrate diagnostic and therapeutic functions, enable real-time monitoring of drug delivery and response, facilitating personalized treatment adjustments [287]. RNA-based therapeutics, such as siRNAs targeting TME immunosuppressive genes (e.g., TGF-β), enhance immunotherapy efficacy, including for exploratory CAR-T-cell therapies limited by TME barriers [285]. Machine learning algorithms can analyze vast datasets of nanocarrier properties and biological interactions to predict optimal designs. As a result, AI algorithms can predict how nanocarriers will behave in the body, including their biodistribution and release kinetics. Integration of AI into nanoparticle design is an evolving field, promising to optimize the nanocarrier size, shape, surface charge, and drug release kinetics for individualized patient applications [288]. Despite these innovations, challenges include scalability, reproducibility, immune clearance (e.g., macrophage uptake), and long-term safety [279,289]. Robust regulatory frameworks and clinical trials are essential for translation. Overall, nanotechnology redefines BC treatment by enhancing specificity, overcoming resistance, and minimizing toxicity. Integrating AI-driven and TME-targeted innovations is likely to pave the way for personalized care.

## 10. Epigenetic Mechanisms Driving Resistance

Epigenetic modifications such as DNA methylation, histone modifications, and non-coding RNAs play pivotal roles in BC progression and therapy resistance by regulating gene expression. One well-documented mechanism is hypermethylation of the ESR1 promoter suppressing estrogen receptor-α (ERα) expression and leading to a poor response to endocrine therapies like tamoxifen [290]. Likewise, aberrant histone modifications such as increased H3K27 trimethylation mediated by EZH2 and reduced H3K27 acetylation silence tumor suppressor genes and have been implicated in therapy resistance. EZH2 overexpression further promotes epithelial–mesenchymal transition, invasion, and drug resistance in BC (Figure 6) [291].

Histone demethylase KDM2A, targeting H3K36me2, is upregulated in aggressive BC subtypes like TNBC. KDM2A enhances cancer stem cell traits and angiogenesis through JAG1 activation and represses tumor suppressors such as E-cadherin by inhibiting TET2. MicroRNAs, notably miR-155, also contribute to resistance; miR-155 can modulate apoptosis and proliferation, although its precise mechanisms in therapy resistance require further elucidation [292]. Epigenetic dysregulation also affects the TME, leading to CD8^+^ T-cell exhaustion and immune evasion [293]. For example, chromatin changes in tumor-infiltrating T-cells drive exhaustion by upregulating immune checkpoint genes, reducing immunotherapy efficacy. Moreover, KDM2A contributes to metabolic rewiring of CSCs via regulation of PGC-1α and promotes stromal remodeling through activation of cancer-associated fibroblasts, further impairing immune and stromal surveillance in BC [293].

The reversible nature of epigenetic alterations presents an attractive avenue for overcoming BC resistance [294,295,296]. DNA methyltransferase inhibitors (DNMTis; azacitidine and decitabine) have demonstrated efficacy in reversing promoter hypermethylation of tumor suppressor genes in preclinical BC models [297]. Histone deacetylase inhibitors (HDACis) such as vorinostat and panobinostat similarly remodel chromatin to enhance tumor immunogenicity and immune-mediated clearance [298].

Emerging evidence suggests that epigenetic agents can re-sensitize endocrine-resistant BC. Recent reports show that decitabine can reconfigure 3D chromatin looping in estrogen-receptor-positive cells, restoring sensitivity to hormonal therapies [299]. Additionally, cancer-derived interleukin-6 (IL-6) activates KDM2A in cancer-associated fibroblasts via the STAT3/NF-κB p50 pathway, promoting stromal-mediated resistance [300]. Refined therapeutic strategies that incorporate epigenetic profiling of tumor and immune compartments can enable biomarker-driven, personalized epigenetic therapy—potentially overcoming BC resistance with enhanced precision (see Figure 6).

## 11. Artificial Intelligence in Breast Cancer Diagnosis and Therapy

Artificial intelligence (AI) is reshaping the landscape of BC research and care, offering transformative capabilities across early detection, diagnostics, treatment planning, and therapeutic innovation. Through the integration of high-dimensional data ranging from medical imaging and pathology to genomics, spatial transcriptomics, and electronic health records, AI empowers a ‘precision oncology framework’ that is adaptive, scalable, and deeply personalized.

In diagnostics, AI excels at pattern recognition, uncovering subtle imaging features often missed by conventional approaches. Deep learning (DL) algorithms outperform traditional radiology in detecting microcalcifications, architectural distortions, and asymmetries in mammography and MRI, especially in dense breast tissue and early-stage tumors [301,302,303,304]. Radiomics-based models can predict the BC risk from screening images and identify aggressive subtypes such as TNBC and HER2-positive disease [303,304]. In digital pathology, DL platforms analyze whole-slide images to quantify tumor-infiltrating lymphocytes (TILs), assess mitotic activity, and classify BC subtypes with high reproducibility. These systems also standardize immunohistochemistry (IHC) scoring for ER, PR, and HER2, enhancing diagnostic consistency [304,305]. Integration with liquid biopsy data, including ctDNA and methylation signatures, further supports non-invasive diagnosis and monitoring [306,307].

The clinical utility of AI is underscored by several recent FDA approvals across oncology. Notably, Ibex Prostate Detect, an AI-powered digital pathology system for prostate biopsy interpretation, and OnQ^TM^; Prostate, an MRI post-processing tool using restriction spectrum imaging (RSI), were granted FDA 510(k) clearance in 2024–2025. In BC, the FDA has cleared two AI-based solutions aimed at improving detection and risk prediction:(1)ProFound AI Detection Version 4.0 (iCAD), a mammography-based AI tool integrating prior exams to boost sensitivity by up to 22%, with reports of a 23% increase in overall cancer detection, a 4% rise in invasive cancer detection, and a doubling of lobular cancer detection. In dense breasts, detection was improved by 32%, with a 40% reduction in T2-stage tumors—all achieved without increasing DCIS detection or recall rates.(2)Clairity Breast, the first AI platform to receive FDA de novo clearance (June 2025), uses routine screening mammograms to predict a patient’s 5-year risk of developing BC, offering high-precision prognostic modeling directly from imaging data (Table 3). These examples illustrate how AI is enhancing diagnostic accuracy, revealing imaging biomarkers imperceptible to the human eye, increasing reproducibility across clinicians, and supporting improved patient stratification—hallmarks of radiology-informed precision oncology.

Beyond detection, AI also plays a growing role in identifying resistance mechanisms and optimizing therapy. Machine learning (ML) models can predict resistance to endocrine therapies or HER2-targeted agents based on genomic alterations (e.g., BRCA1/2, PIK3CA) and tumor microenvironmental factors [308]. Immune phenotyping algorithms help identify immune evasion signatures and stratify patients for checkpoint inhibitors, particularly in immunologically ‘cold’ TNBC subtypes [309]. AI is also revolutionizing nanomedicine. By refining nanoparticle architecture for improved targeting, AI reduces off-target toxicity and enhances drug delivery to challenging microenvironments such as hypoxic or fibrotic tumors [310,311]. These advances support the development of customized therapies, including antibody–drug conjugates (ADCs) and endocrine regimens guided by real-time molecular profiling [312]. In drug development, AI accelerates discovery by mapping synthetic lethality (e.g., BRCA/PARP), simulating protein folding (e.g., AlphaFold), and optimizing synergistic combinations such as PI3K inhibitors in hormone-receptor-positive BC [308,313].

Despite the momentum, significant barriers remain. Data heterogeneity, limited generalizability, privacy concerns, and lack of standardized multi-institutional datasets challenge model deployment [16,314]. Large language models (LLMs), including ChatGPT-3 and ChatGPT-4o, show promise in oncology education and decision support but currently lack rigorous domain-specific calibration and often produce inconsistent or hallucinated outputs [314,315].

Future innovation will depend on (1) federated learning, enabling decentralized AI training across institutions while preserving patient privacy [316]; (2) explainable AI (XAI), promoting interpretability and clinical trust [181]; (3) digital twins, simulating tumor evolution and treatment response in silico [317]; and (4) EHR-integrated AI, for real-time prediction and clinical decision-making (Figure 7) [317]. Together, these developments suggest that AI is not simply an adjunct but a core driver of precision oncology, capable of evolving with tumor biology and personalizing cancer care in unprecedented ways.

Artificial intelligence (AI) integrates seamlessly into breast cancer research by unifying resistance biology with diagnostic and therapeutic innovation [302,307]. Beyond enhancing imaging and pathology, AI connects core mechanisms of resistance, including drug efflux, genetic and epigenetic alterations, and metabolic–immune reprogramming. It enables prediction of the therapy response, monitoring of disease evolution through circulating tumor DNA (ctDNA), and stratification of high-risk subgroups such as TNBC patients [62,313,318]. By modeling apoptotic and other regulated cell death pathways, AI also reveals new therapeutic vulnerabilities [270]. In this way, AI serves not as an adjunct but as a catalyst—bridging molecular insights, resistance mechanisms, and clinical decision-making into a coherent framework for overcoming therapy resistance in breast cancer [319].

## 12. Personalized Medicine in Overcoming Resistance

Personalized medicine is redefining BC therapy by aligning treatment strategies with the unique molecular profile of each tumor. By integrating genomic, proteomic, and metabolomic data, clinicians can identify resistance-driving mutations—such as *ESR1* in HR^+^ disease or *BRCA* in TNBC—and tailor interventions like elacestrant or PARP inhibitors accordingly [68,72]. Beyond single-gene targeting, multi-omics approaches support rational drug combinations and pathway-specific inhibitors, including PI3K-targeted agents in *PIK3CA*-mutant tumors [71]. Emerging modalities—from nanocarrier-enhanced delivery systems and exosome-based interventions to CAR-T-cell therapies and macrophage reprogramming—aim to overcome barriers posed by tumor heterogeneity and an immunosuppressive microenvironment [236,300]. When layered with AI-driven analytics, this evolving landscape offers a multidimensional framework to counter therapeutic resistance and advance precision oncology in BC care [181,315,318].

## 13. Conclusions

Despite advances in early detection and targeted therapies, treatment resistance remains a defining obstacle in breast cancer management, particularly in aggressive subtypes such as TNBC and metastatic disease. Overcoming resistance will require strategies that are adaptable, multidimensional, and firmly grounded in translational frameworks.

Resistance arises from the convergence of tumor heterogeneity, metabolic adaptation, immune evasion, and dynamic remodeling of the tumor microenvironment (TME). Progress in molecular oncology and biomedical engineering is opening up new avenues to address these challenges. Artificial intelligence (AI) should be regarded not as a separate tool but as an integrative connector that links early detection through ctDNA and imaging, prediction of efflux and genetic or epigenetic resistance, modeling of cell death pathways, and stratification of high-risk subgroups such as TNBC. Recent FDA-approved AI screening modules already in clinical use illustrate this potential, and AI continues to advance the integration of multi-omics datasets, radiomic signatures, and rational drug design, enabling patient-specific therapeutic strategies. In parallel, nanomedicine, exosome-based delivery platforms, and TME-directed immunotherapies are expanding the therapeutic arsenal against resistant breast cancer.

This review underscores four areas of particular translational significance. First, experimental models that replicate the complexity of resistance biology have improved mechanistic understanding and drug screening. Second, tumor vasculature functions not only as a nutrient conduit but also as a structural and metabolic regulator of the TME. Third, metabolic rewiring within cancer and immune cells shapes disease progression and treatment outcomes. Finally, the breast tissue microbiome is emerging as both a contributor to carcinogenesis and a mediator of therapeutic response. Closing the gap between these mechanistic insights and clinical application will require interdisciplinary collaboration, system-level integration, and rigorous validation. Aligning molecular oncology, strategies targeting the metabolic–immune axis, and bioengineering within patient-centered frameworks will be essential for transforming resistance from an inevitable obstacle into a surmountable challenge.

### Future Directions

In future, the next phase of breast cancer research should shift from reacting to resistance toward anticipating and preempting it. Precision mapping of resistance at a single-cell resolution, combined with spatial transcriptomics, can expose the unique vulnerabilities of resistant subclones. Integrating these insights with genomic, proteomic, and metabolomic profiles will allow treatment plans to be tailored with unprecedented specificity, moving beyond broadly applied regimens toward strategies that match the biological profile of each patient.

The analysis of the data from publications revealed the pathways and also specific components of the pathways that are responsible for the resistance of BC. The executioners of the pathways are proteins and RNA in some cases. These include receptors, ion channels and enzymes. Hence, both the involved proteins and RNA may serve as ideal targets for the development of drugs. Such an approach is viable as there are FDA-approved drugs available on the market targeting the cellular proteins and RNA. These include KRAS inhibitors, proteosome inhibitors, monoclonal antibodies and RNAi. It is likely that the targeting of a common enzyme or receptor may lead to undesirable side effects in healthy cells besides cancer cell.

Based on this, the evaluation of candidate drugs should be carried out using appropriate experimental model systems. Immunotherapy must be adapted to overcome the suppressive influence of the TME. This will require combining next-generation immune effectors, such as CAR-T and CAR-NK cells, with TME-reprogramming strategies that sustain immune activation. Advances in regulated cell death biology offer additional opportunities to selectively eliminate resistant tumor cells. The emerging liquid-biopsy-based technologies for detecting circulating tumor DNA, tumor-associated autoantibodies, and exosomal markers must be advanced as early-warning systems for resistance. Such tools will enable continuous disease monitoring, guiding timely therapeutic adjustments.

Artificial intelligence will remain pivotal, not only in diagnostics but also in high-throughput drug discovery, nanoparticle design, and adaptive treatment modeling. The convergence of mechanistic insight, engineering innovation, and computational power offers a realistic path where sustained control and cure are achievable outcomes in BC.

## Figures and Tables

**Figure 1 cancers-17-02938-f001:**
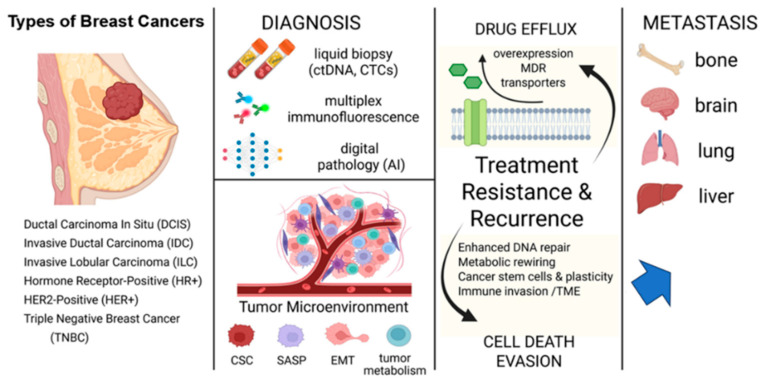
Overview of the pathogenesis of breast cancer. This figure illustrates the complexity of the disease progression through the stages and their classification, the role of the TME in metastasis and a short list of resistant mechanisms.

**Figure 2 cancers-17-02938-f002:**
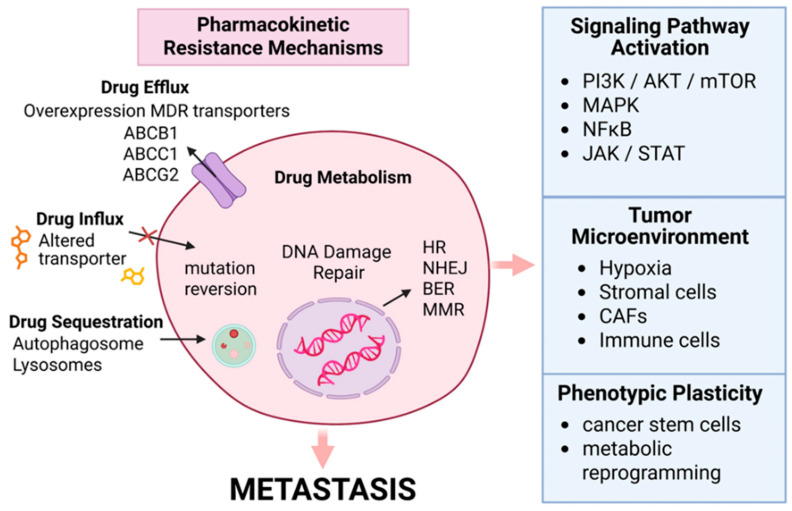
Key molecular signaling pathways involved in BC that contribute to therapeutic resistance mechanisms.

**Figure 3 cancers-17-02938-f003:**
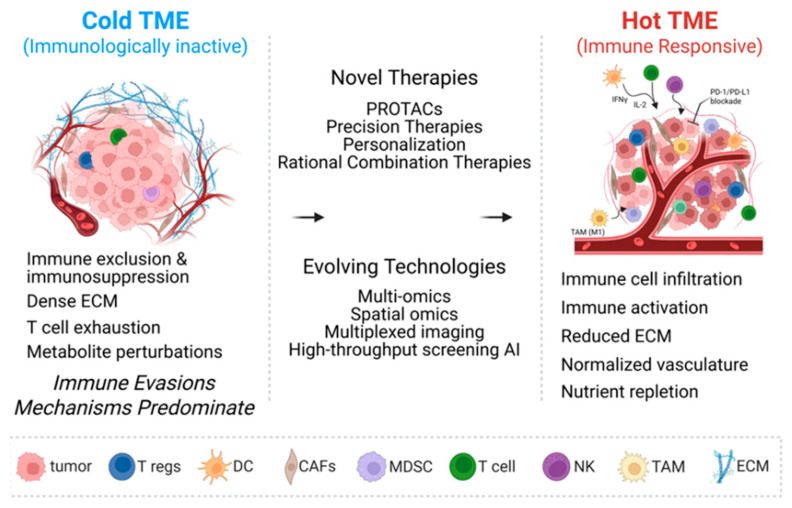
Tumor microenvironment (TME) modulation: converting ‘cold’ to ‘hot’. Simplified depiction of the transition from an immunologically inert (‘cold’) to an immune-responsive (‘hot’) tumor microenvironment, highlighting therapeutic strategies aimed at overcoming immune exclusion and enhancing antitumor responses.

**Figure 4 cancers-17-02938-f004:**
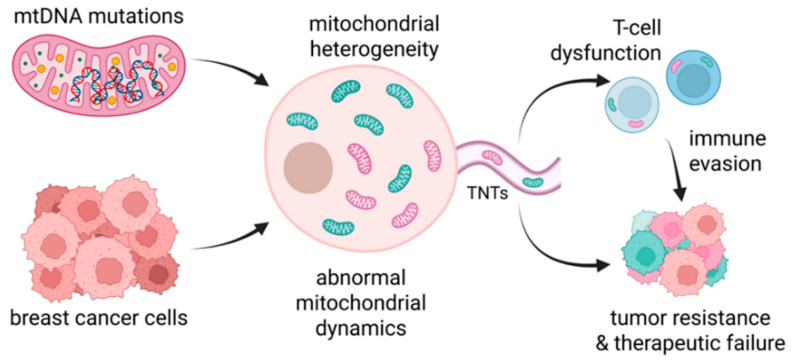
Mitochondrial dynamics and intercellular transfer drive immune evasion and tumor resistance in breast cancer. BC cells exhibit mitochondrial DNA mutations, heterogeneity, and abnormal mitochondrial dynamics. Through tunneling nanotubes (TNTs), these cells transfer dysfunctional mitochondria to CD4^+^ and CD8^+^ T-cells, leading to T-cell dysfunction and immune evasion.

**Figure 5 cancers-17-02938-f005:**
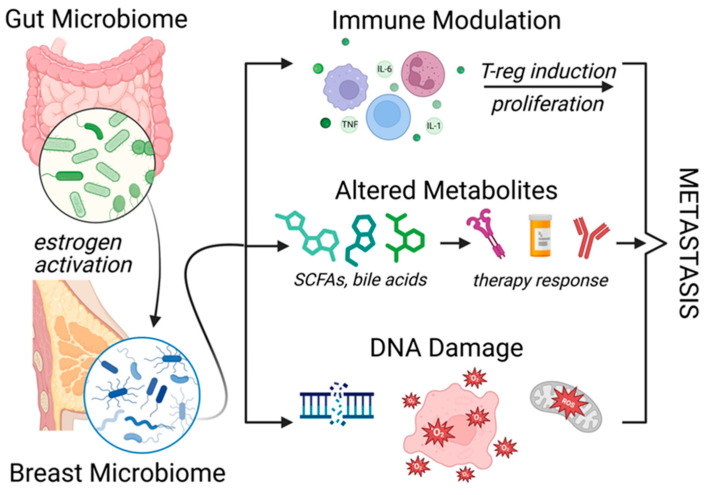
Microbiota-driven mechanisms of BC resistance and progression. The gut and breast tissue microbiomes, including chemotherapy-induced alterations, shape tumor biology. This occurs through estrogen reactivation, DNA damage (colibactin, ROS), immune modulation (IL-6, TNFα, IL-1, Treg induction), and microbial metabolites (SCFAs, bile acids).

**Figure 6 cancers-17-02938-f006:**
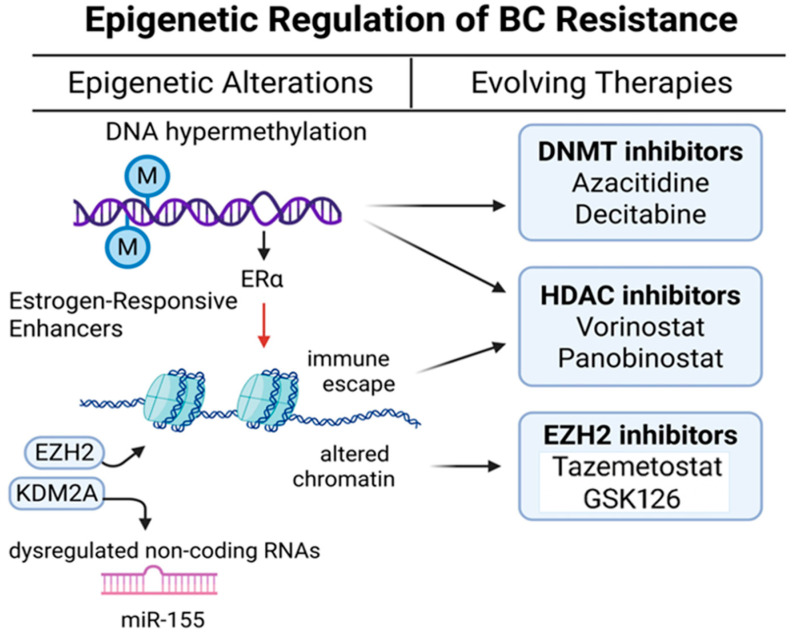
Epigenetic mechanisms contributing to BC resistance and targeted therapies. DNA hypermethylation, histone modifications (EZH2, KDM2A), and dysregulated non-coding RNAs (e.g., miR-155) promote ERα silencing, chromatin remodeling, and immune escape. Targeted therapies include DNMT inhibitors, HDAC inhibitors, and EZH2 inhibitors aimed at reversing these resistance mechanisms.

**Figure 7 cancers-17-02938-f007:**
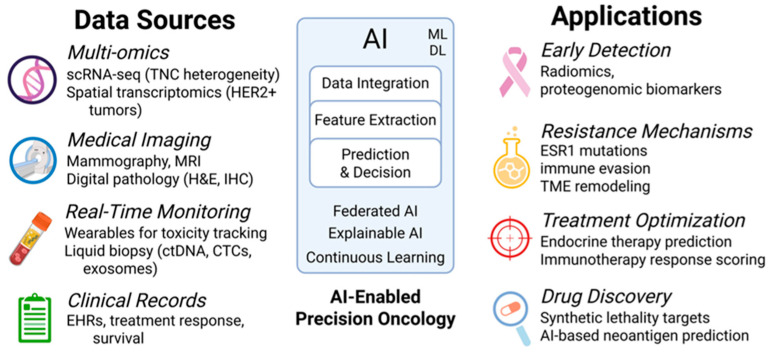
AI-enabled precision oncology in breast cancer. Multi-modal inputs are laid out in the left column, AI-technology in the middle and the applications on the right side. Taken together, a logical transition from data collection to analysis and the end result are laid out logically and cohesively.

**Table 1 cancers-17-02938-t001:** Bioinformatics approach to therapeutic resistance. Key datasets and associated pathway alterations are summarized; supporting references are provided in the main text.

Dataset	Altered Pathways/Features	Resistance Mechanisms
TCGA, METABRIC	BRCA1, BRCA2, TP53, PIK3CA, ESR1 mutations	Drug efflux, Immune evasion
GEO, ArrayExpress	Proliferation (Ki-67, CCND1);	Immune evasion
	Immune evasion (PD-L1/CD274, CTLA4);	
	Metabolic rewiring (HK2, LDHA)	
CPTAC (Proteogenomic)	PI3K/AKT/mTOR,	Immune evasion
	MAPK signaling;	
	DNA repair networks	
Epigenomic datasets	Promoter hypermethylation (ESR1, BRCA1);	Immune evasion, EMT, Stemness
	ncRNAs (miR-21, HOTAIR)	
Meta-analyses (cBioPortal, KM-Plotter)	Stemness (ALDH1A1, SOX9);	Stemness
	EMT drivers (TWIST1, SNAI2)	

**Table 2 cancers-17-02938-t002:** Experimental models in breast cancer resistance research.

Model	Key Contributions	Pros	Cons	Ref
2D Cell Lines	Mechanistic discoveries in ER, HER2, and efflux resistance	Cheap, reproducible, high-throughput	Poor physiological relevance	[67,68,69]
3D Spheroids/Organoids	Hypoxia, CSC-driven resistance, TME influence	Mimics architecture, patient-derived	Complex culture, batch variability	[67,68,74]
Patient-Derived Xenografts (PDX)	Resistance in heterogeneous tumors, therapy validation	High translational relevance	Expensive, lacks human immunity	[68,74,75]
Genetically Engineered Mice (GEMMs)	DNA repair defects, immune-competent resistance models	Immune-competent, spontaneous tumors	Genetically rigid, costly	[68,75]
CTC Models	Insights into metastasis, mesenchymal resistance	Real-time, metastatic focus	Difficult to culture and expand	[75,76]
In Silico Models (AI-based)	Predictive modeling of resistance, target discovery	Fast, scalable, cost-effective	Needs biological validation	[68]

**Table 3 cancers-17-02938-t003:** Next-generation FDA-cleared AI solutions in cancer diagnostics.

Feature	Ibex Prostate Pathology	OnQ^TM^ Prostate Imaging	AI in Breast Cancer (ProFound 4.0)	AI in Breast Cancer (Clairity Breast)
FDA Status	510(k)-cleared (May 2024)	510(k)-cleared (Feb 2025)	510(k)-cleared (Nov 2024)	De novo clearance (June 2025)
Modality	AI-based analysis of H&E-stained biopsy slides	RSI-enhanced diffusion-weighted MRI	Mammography (with or without prior imaging)	AI-based analysis of screening mammograms
Purpose	Digital pathology interpretation, cancer detection	Improved lesion characterization, biopsy targeting	Enhanced sensitivity and risk prediction	Detection of subtle imaging features predictive of future cancer
Clinical Utility	Gleason scoring, decision support for pathologists	Improves PI-RADS accuracy, reduces inter-reader variability	Improves detection in dense breasts, risk stratification	Predicts 5-year BC risk from routine mammography

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
