# Peer review of "The Underlying Mechanisms and Emerging Strategies to Overcome Resistance in Breast Cancer"

_cancers, 2025, doi:10.3390/cancers17172938_

Round 1
Reviewer 1 Report
Comments and Suggestions for Authors
This review article submitted to Cancers J by MDPI, titled “The Underlying Mechanisms and Emerging Strategies to Over- 2
come Resistance in Breast Cancer” by Kannan et al., 2025
- In the current research the authors addressed the biological mechanisms causing BC resistance and its potential treatment. AI's potential to address critical challenges transforming BC therapy for improved survival rate and quality of life for the patients, were addressed in the current review as well.
- The topic is original and relevant to the j and the field
The conclusions is consistent with the evidence and arguments presented
and address the main question posed, but avoid extrapolation of the findings, please rewrite and restructure.
Minor corrections:
- The abstract needs more details and better to be structured,
- keywords: need more words
- Several sentences are long and without ref.
- Please split long sentences and add a ref. for each single sentence or each single info.
- To the short aim please better write the search strategy (SS)
- Line 159 life style not lifestyle
- Add future directions
Major corrections:
- Figure 1, and all other figures, with no addition to the knowledge, these are known and shown in previous papers, please innovate and add new info to each and btter to be summary to the text, enrich all figures more, they are shallow,
- Therapeutic strategies and all coming title heading needs subheading for each medication or each info below in other sections,
- Please organize btter
- Resistance mechanisms in BC needs subheadings for the info below,
- What about the triple negative type
- What about the bone affection in BC
- Enumerate the BC prognosis indices ?
- Table 1 is not based on bioinformatics bases only on literature, please include bioinformatics
- Add a Graphical abstract
- List of abbreviations to be added
Author Response
Authors thank the reviewers for their time and constructive suggestions on the manuscript. We have carefully read the comments and addressed the concerns raised by the reviewers. The manuscript has been revised in accordance with their suggestions. The revision includes the changes in the text, figures and Tables. The revised text is highlighted in red. Our detailed responses are presented below in bold text, under each comment of the reviewers (indicated by italics).
- In the current research the authors addressed the biological mechanisms causing BC resistance and its potential treatment. AI's potential to address critical challenges transforming BC therapy for improved survival rate and quality of life for the patients, were addressed in the current review as well.
- The topic is original and relevant to the j and the field
- The conclusions is consistent with the evidence and arguments presented and address the main question posed, but avoid extrapolation of the findings, please rewrite and restructure.
We thank the reviewer for the overall positive comments on our manuscript.
Minor corrections:
- The abstract needs more details and better to be structured.
Agree. Accordingly, we have restructured our abstract and included detailed information on some of the mechanisms outlined in the manuscript.
- keywords: need more words
Agree. Added new keywords.
- Several sentences are long and without ref.
Agree. Revised the entire manuscript for structure, grammar, and better readability.
Long sentences have been made shorter and easy to read. Wherever needed, additional references have been included.
- Please split long sentences and add a ref. for each single sentence or each single info.
Agree. Revised our text accordingly. Long sentences were spliced into smaller and readability has been improved throughout the manuscript. Additional references were added where appropriate.
- To the short aim please better write the search strategy (SS)
Agree. A search strategy to obtain literature used in the manuscript was added in section 1.1.
- Line 159 life style not lifestyle
On the use of ‘lifestyle’ vs ‘life style’, we respectfully disagree with the reviewer since the word ‘lifestyle’ is what routinely used nowadays.
- Add future directions
Agree. Concluding remarks have been revised and future directions have been added.
Major corrections:
- Figure 1, and all other figures, with no addition to the knowledge, these are known and shown in previous papers, please innovate and add new info to each and better to be summary to the text, enrich all figures more, they are shallow.
All the figures included in the manuscript have been updated with latest information in the literature. The text in the manuscript was revised accordingly. Few figures that did not significantly added to the knowledge base were deleted. The text was updated accordingly in the manuscript.
The updated details are:
> Figure 1: Modified with updated information.
> Figure 2: This figure was deleted since Table 2 had similar details. Thus, we avoided duplications and retained Table 2.
> Figure 3: Updated
> Figure 4: Updated
> Figure 5: Updated
> Figure 6: Deleted
> Figure 8: Deleted
> Figure 10: Updated
> A new Graphic abstract figure was added.
> New Figure numbers were revised in the order in which they appear.
> In the case of Tables, Table 1 and 2 were deleted.
> A new table on Bioinformatics included.
- Therapeutic strategies and all coming title heading needs subheading for each medication or each info below in other sections,
Agree. Now, subheadings have been added on any new information wherever appropriate.
- Please organize better
We have reorganized portions of the text or improved readability and flow. We have added graphical abstract and deleted a few figures to make the review concise and sharp. We have also added sections on triple negative breast cancer (TNBC) and bone-related pathology as relevant to breast cancer metastasis. Regulated cell death mechanisms (apoptosis, immunogenic cell death, pyroptosis, ferroptosis etc.) were included in a separate section to better explain resistance mechanisms.
- Resistance mechanisms in BC needs subheadings for the info below,
Agree. Revised the manuscript with appropriate subheadings.
- What about the triple negative type
Agree. Our original manuscript had scattered mentions of TNBC across the text. We now have revised and added a separate section on TNBC.
- What about the bone affection in BC
Agree. Added new section on bone pathology.
- Enumerate the BC prognosis indices?
The primary aim of this review was to understand the cellular and molecular mechanisms of resistance in BC. Furthermore, we focused on understanding multiple resistance mechanisms in breast cancer and emerging therapeutic strategies to overcome. The review does not focus or analyze any clinical indices of BC development or it’s prognosis from clinical point of view. Our review was more focused on mechanisms, therapeutic developments and emerging technologies such as the role of AI. Therefore, we did not develop a separate section on ‘Enumerate the BC prognosis indices.’
- Table 1 is not based on bioinformatics bases only on literature, please include bioinformatics
Agree. Table 1 on life style factors influencing BC has now been deleted and replaced with a new Table on Bioinformatics based approach to BC.
- Add a Graphical abstract
Agree. A new Graphical abstract is added.
- List of abbreviations to be added
Agree. List of Abbreviations have been added.
Reviewer 2 Report
Comments and Suggestions for Authors
The article entitled “The Underlying Mechanisms and Emerging Strategies to Overcome Resistance in Breast Cancer” is very interesting, as it provides a comprehensive overview of the major mechanisms underlying resistance in breast cancer. The manuscript also commendably includes emerging approaches such as the application of modern technologies, including AI, for overcoming resistance.
However, some minor revisions are required:
-
Introduction – The introduction should be more concise, as it is currently overly detailed.
-
Resistance to Apoptosis – This is a major contributor to therapy failure in breast cancer and should be included in the discussion.
-
Resistance Due to Drug Efflux – This section should provide more detail on relevant transporters, such as Breast Cancer Resistance Protein (BCRP), and should also include strategies to overcome multidrug resistance (MDR) in breast cancer.
Author Response
Authors thank the reviewers for their time and constructive suggestions on the manuscript. We have carefully read the comments and addressed the concerns raised by the reviewers. The manuscript has been revised in accordance with their suggestions. The revision includes the changes in the text, figures and Tables. The revised text is highlighted in red. Our detailed responses are presented below in bold text, under each comment of the reviewers (indicated by italics).
The article entitled “The Underlying Mechanisms and Emerging Strategies to Overcome Resistance in Breast Cancer” is very interesting, as it provides a comprehensive overview of the major mechanisms underlying resistance in breast cancer. The manuscript also commendably includes emerging approaches such as the application of modern technologies, including AI, for overcoming resistance.
We thank the reviewer for the overall positive comment on our manuscript.
However, some minor revisions are required:
- Introduction – The introduction should be more concise, as it is currently overly detailed.
Agree. Introduction has been revised and made concise. Changes are highlighted in red.
- Resistance to Apoptosis – This is a major contributor to therapy failure in breast cancer and should be included in the discussion.
Agree. A new section on apoptosis (section 4.5. Evading Apoptosis in Breast Cancer Resistance) has been added with relevant information on various apoptotic mechanism and how they evade therapeutic treatment.
- Resistance Due to Drug Efflux – This section should provide more detail on relevant transporters, such as Breast Cancer Resistance Protein (BCRP), and should also include strategies to overcome multidrug resistance (MDR) in breast cancer.
Agree. We have revised the section 4.2 (Resistance due to Drug Efflux) to include Breast Cancer Resistance Proteins (BRCP/ABCG2) in relevance to drug resistance. Figure 2 in our review also depicts ABCG2(BRCP) and other transporters associated resistance mechanisms.
Reviewer 3 Report
Comments and Suggestions for Authors
The present review by Kannan et al about the emerging strategies to overcome resistance in breast cancer, indicating in the abstract that they will explore the role of AI, at the beginning was very interesting. But a profound analysis show several deficiencies:
1) Any review analysis requires and a PICO approach and show a PRISMA flow diagram that depicts the flow of information through the different phases of a systematic review. Which are absent in this review.
2) The review show a lot of information of several different aspects of breast cancer, without going into any of them in depth, being rather informative and, therefore, its contribution to the field is faint.
3) Is not evident the research field of authors. A good starting point for a review, focus in one or two aspects, is the main research field of authors. Because this provides a basis for the review, allowing it to focus on few but important aspects of the field of research and not try to be, excuse the phrase, a kind of textbook chapter.
4) Authors indicate in summary the important role of AI in this field, however in text, they only list some AI developments without going into depth on them.
Therefore, at present manuscript is not suitable for publication and requires major changes, focus in the reduction of the review topics using his own research as a starting point (as reccomendation)
Author Response
Authors thank the reviewers for their time and constructive suggestions on the manuscript. We have carefully read the comments and addressed the concerns raised by the reviewers. The manuscript has been revised in accordance with their suggestions. The revision includes the changes in the text, figures and Tables. The revised text is highlighted in red. Our detailed responses are presented below in bold text, under each comment of the reviewers (indicated by italics).
The present review by Kannan et al about the emerging strategies to overcome resistance in breast cancer, indicating in the abstract that they will explore the role of AI, at the beginning was very interesting. But a profound analysis show several deficiencies:
- Any review analysis requires and a PICO approach and show a PRISMA flow diagram that depicts the flow of information through the different phases of a systematic review. Which are absent in this review.
We thank the reviewer for his/her constructive comments. The goal of our review was to analyze the complex cellular and molecular mechanisms of BC resistance. This exercise revealed both unique and shared mechanisms contributing to resistance pathways.
We have endeavored to meet the goals by a systematic review of literature by utilizing PubMed, Scopus, Web of Science, EMBASE, and Cochrane Library (data bases) covering peer-reviewed. Further, this was supplemented by manual searches and cross-referencing of medical journals. Boolean operators (AND, OR, NOT) combined with keywords and controlled vocabularies (e.g., MeSH, EMTREE) were used to ensure a comprehensive retrieval.
The complexity of the BC subtypes and their heterogeneity required multiple approaches to retrieve the literature. All co-authors were involved in screening and assessing full-text peer-reviewed articles.
The methodology employed in literature search is outlined in the introductory section 1.1 of this review article.
- The review show a lot of information of several different aspects of breast cancer, without going into any of them in depth, being rather informative and, therefore, its contribution to the field is faint.
We respectfully disagree. The review article was written in response to a special invitation from the journal ‘Cancers’ following approval of the topic and its scope. This review synthesizes the extensive literature on resistance mechanisms and therapeutic strategies, providing an in-depth understanding of their molecular complexities.
We are aware that reviews emphasizing a specific mechanism are available. However, our approach here is to present a consolidated review highlighting multiple mechanisms and provide a broader overview. Such consolidated presentations have been valuable to us, and we hope that this presentation is similarly valuable to others.
- Is not evident the research field of authors. A good starting point for a review, focus in one or two aspects, is the main research field of authors. Because this provides a basis for the review, allowing it to focus on few but important aspects of the field of research and not try to be, excuse the phrase, a kind of textbook chapter.
We respectfully disagree with the reviewer’s opinion about the author’s expertise in cancer research and publications thereof. All authors have been involved in cancer research and have published in peer-reviewed journals.
- Authors indicate in summary the important role of AI in this field, however in text, they only list some AI developments without going into depth on them.
Our revised review article now covers all fundamental aspects of evolving AI. We have also revised the figure to reflect the advancements in AI including medical imaging, real-time monitoring, treatment optimization, drug discovery etc.
- Therefore, at present manuscript is not suitable for publication and requires major changes, focus in the reduction of the review topics using his own research as a starting point (as reccomendation)
This manuscript has been extensively revised based on the constructive and scientific criticisms by the reviewers. Those criticisms have helped to improve the quality of the review.
Round 2
Reviewer 1 Report
Comments and Suggestions for Authors
Still the manuscript needs to be improved
Author Response
Agree. As suggested by the reviewer 1, we have revised the manuscript thoroughly to improve the quality for publication. Revisions include changes in the text, updating figures, addition of new information with references, organization of headings and subheading to improve the flow of information.
Reviewer 3 Report
Comments and Suggestions for Authors
I really appreciate the work made by authors in order to improves the manuscript. The suggestion made were accepted and are clearly indicated in this new version.
I agree with the explanations about some of my concerns and after read the new version my decicson is accept the review for publication
Congratulations to all the team
Author Response
Authors thank the reviewer 3 for positive comments and for recommending the review article for publication in its present form.